# LARGE-SCALE PUBLIC DATA IMPROVES DIFFERENTIALLY PRIVATE IMAGE GENERATION QUALITY

## ABSTRACT

Public data has been frequently used to improve the privacy-accuracy trade-off of differentially private machine learning, but prior work largely assumes that this data come from the same distribution as the private. In this work, we look at how to use *generic* large-scale public data to improve the quality of differentially private image generation in Generative Adversarial Networks (GANs), and provide an improved method that uses public data effectively. Our method works under the assumption that the support of the public data distribution contains the support of the private; an example of this is when the public data come from a general-purpose internet-scale image source, while the private data consist of images of a specific type. Detailed evaluations show that our method achieves SOTA in terms of FID score and other metrics compared with existing methods that use public data, and can generate high-quality, photo-realistic images in a differentially private manner.

## 1 INTRODUCTION

Differential privacy (DP) (Dwork et al., 2006; 2014) is considered the gold standard for privacy in machine learning and data analytics with sensitive data, with many use-cases in industry and government. While differentially private machine learning has seen considerable recent progress (Li et al., 2021; Yu et al., 2021; De et al., 2022), the main challenge remains balancing the privacy-utility trade-off. DP provides individual-level privacy by injecting noise into the training process in order to obscure the private value of a single data point. This reduces the statistical efficiency, or accuracy per sample of the trained model – sometimes rather significantly. As a result, a large body of work in differentially private machine learning has focused on how to design algorithms that provide better privacy vs. statistical efficiency trade-offs (Abadi et al., 2016; Sheffet, 2017; Iyengar et al., 2019).

In particular, for use-cases that involve small amounts of private data, a line of work has looked into combining sensitive data with publicly available data to improve the utility of private machine learning (Papernot et al., 2016; 2018; Zhu et al., 2020). However, the vast majority of this line of work assume that the public data is drawn from the same data distribution as the private data. This assumption is unrealistic in practice, since very often public data comes from a different source and may have very different qualities than the private data.

In this work, we relax this assumption and consider the problem of generative modeling of a distribution of private images based on *generic* large-scale public data. In particular, we assume that the support of the public data distribution contains the support of the private. An example use-case is when the public data comes from a general-purpose internet-scale source (such as ImageNet (Deng et al., 2009)), while the private data consists of images of a specific type. Under this assumption, our goal is to train a Generative Adversarial Network (GAN) to generate samples from the private data distribution while preserving differential privacy of the private data.

The main challenge in this problem is ensuring high image generation quality. Learning the private distribution either through direct training or fine-tuning using differentially private stochastic gradient descent (DP-SGD; (Song et al., 2013; Abadi et al., 2016)) requires the addition of a large amount of noise during training, which results in noisy models that generate blurry or malformed images. The key insight in our work is that instead of privately learning to generate highly detailed images from scratch, it is much more efficient to privately *adapt* a generative model trained on public data.

To leverage this insight, we first use a pair of encoder and decoder trained entirely on public data. The encoder maps images to a low-dimensional feature space and the decoder generates images given a feature vector. This architecture effectively reduces the problem of learning the private image distribution to learning a private *feature* distribution in the latent space of the encoder. We do so either by modeling the private features as a multivariate Gaussian, or by modeling the difference between the public and private feature distributions using a density ratio estimator. Both methods are highly sample-efficient when applied differentially privately. Using the estimated private feature distribution, we then sample from it to obtain feature vectors and use the decoder to generate a new image from the private image distribution.

Finally, we evaluate our proposed algorithms choosing ImageNet as public data, and six separate image datasets as private data. We show that when privacy levels are moderate to high, our algorithms vastly outperform existing baselines in terms of FID scores as well as other distribution quality metrics. Visual inspection of the generated images reveals that unlike prior work, our methods are capable of producing drawn from the private distribution that are high quality and realistic even for moderate to high privacy levels.

## 2 PRELIMINARIES

### 2.1 DIFFERENTIALLY PRIVATE MACHINE LEARNING

Differential privacy (DP) (Dwork et al., 2006; 2014) is a cryptographically motivated privacy definition and now considered the gold standard in private data analysis. DP applies to a randomized algorithm, and the main idea is that the participation of a single data point in the dataset should not change the probability of any outcome by much. Formally, the definition is as follows.

**Definition 1** (($\varepsilon, \delta$)-Differential Privacy). *Let $\varepsilon, \delta \in \mathbb{R}^{\geq 0}$. A randomized algorithm $\mathcal{M} : (\mathcal{X} \times \mathcal{Y})^n \to \mathcal{R}$ with domain $(\mathcal{X} \times \mathcal{Y})^n$ and range $\mathcal{R}$ satisfies ($\varepsilon, \delta$)-differential privacy if for any two datasets $D, D' \in (\mathcal{X} \times \mathcal{Y})^n$ that differ by a single person's private data $(\mathbf{x}, y)$, and for any subset of outputs $S \subseteq \mathcal{R}$, we have:*

$$\mathbb{P}[\mathcal{M}(D) \in S] \leq e^{\varepsilon} \cdot \mathbb{P}[\mathcal{M}(D') \in S] + \delta.$$

Observe that the definition involves two privacy parameters $\varepsilon$ and $\delta$; for both, higher values imply lower privacy. DP has achieved considerable popularity in the literature because of its excellent properties – resistance to prior information, effectiveness against privacy attacks (Yeom et al., 2018; Humphries et al., 2020; Guo et al., 2022), as well as graceful composition under data re-use.

The standard tool for differentially private deep learning is differentially private stochastic gradient descent (DP-SGD; (Song et al., 2013; Abadi et al., 2016)), which aims to train a deep learning model by minimizing an empirical loss function calculated over the training data points. For this purpose, in each iteration, DP-SGD samples a batch of training data points with the poisson sample rate $q$, and calculates the gradients of the loss function corresponding to those points. Each gradient is then clipped to a pre-set norm $C$, and Gaussian noise is added to the average gradient as follows:

$$\hat{g} = \frac{1}{B} \sum_{i=1}^{B} \left( \frac{g_i}{\max(1, \|g_i\|/C)} + \mathcal{N}(\mathbf{0}, \sigma^2 C^2 I) \right),$$

where $g_i$ is the gradient of the loss function corresponding to example $i$ in the batch. Mironov et al. (2019) proposes an advanced privacy accounting method, which can calculate the privacy parameters from the total iterations $T$, poisson sample rate $q$, training set size $n$, and the scale of the noise $\sigma$. Given the privacy parameters ($\varepsilon, \delta$), although there is no explicit form to set the value $\sigma$, a binary search can help find an appropriate $\sigma$ as $\varepsilon$ (with fixed $\delta$) are monotonically increasing as $\sigma$ decreases.

### 2.2 GENERATIVE ADVERSARIAL NETWORKS

For image generation models, we use Generative Adversarial Networks (GANs; (Goodfellow et al., 2020)), a standard tool in deep generative modeling. To facilitate learning from off-distribution data, we use as our backbone the recently proposed Instance-Conditioned GAN (IC-GAN; (Casanova et al., 2021)). IC-GAN has shown excellent performance in generating "transfer" samples.

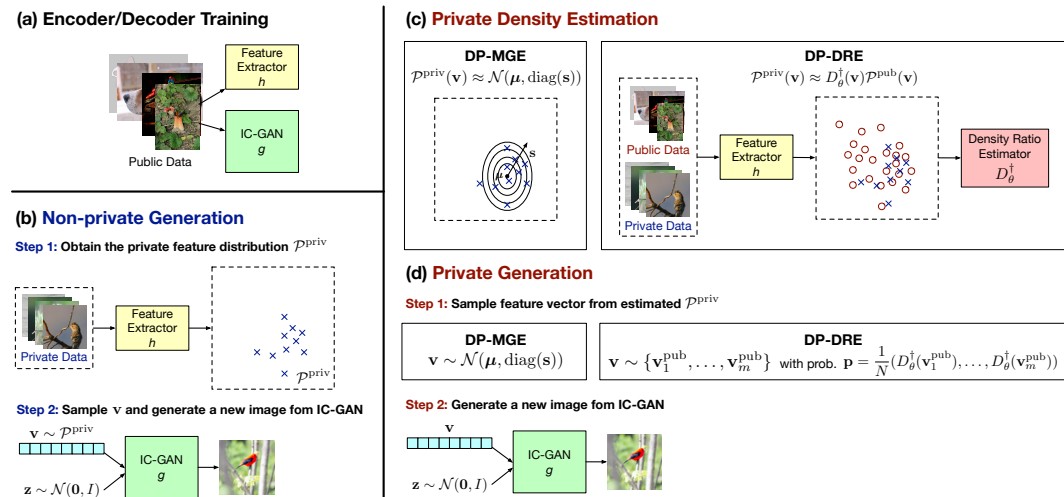

Figure 1: **Method overview.** We first train a feature extractor and IC-GAN on public data in (a), which can be used for non-private data generation from the private distribution $\mathcal{P}^{\mathrm{priv}}$ as shown in (b). To make this generation process private, we estimate $\mathcal{P}^{\mathrm{priv}}$ privately in (c) using either a multivariate Gaussian distribution (DP-MGE; Section 3.1) or a density ratio estimator combined with the public distribution (DP-DRE; Section 3.2). The private generation process then proceeds by sampling a feature vector $\mathbf{v}$ from the estimated distribution and using the IC-GAN to generate an image in (d).

In particular, IC-GAN works as follows. First, we train an unsupervised feature extractor $h$ on the training data, and use it to obtain representations $h(\mathbf{x})$ of the input. Then, we learn a generator that models the data distribution as a mixture of the instance-conditional distributions. Specifically, given an instance $\mathbf{x}$, the IC-GAN generator takes its feature vector $h(\mathbf{x})$ as input and samples from a distribution of images $\mathbf{x}'$ whose feature vectors $h(\mathbf{x}')$ are close to the input feature $h(\mathbf{x})$.

To generate samples from a distribution that is different from the training, we simply need to sample an instance $\mathbf{x}$ from the dataset, compute the feature vector $h(\mathbf{x})$, and generate a new sample by feeding $h(\mathbf{x})$ to the generator. If $\mathbf{x}$ still lies within the support of the training distribution – even if it is not directly drawn from it – the IC-GAN should be able to generate a sample close to it.

## 2.3 PROBLEM SET-UP.

**Problem statement.** Our precise problem statement is as follows. We are given a private image dataset $D^{\mathrm{priv}}$ and an auxiliary public image dataset $D^{\mathrm{pub}}$. Our goal is to learn a generator $G$ that can generate new images such that (a) the distribution of the generated images is close to the distribution of $D^{\mathrm{priv}}$ and (b) the learning algorithm that outputs the generator satisfies $(\varepsilon, \delta)$-differential privacy with respect to $D^{\mathrm{priv}}$. This problem setting follows Harder et al. (2022), and is motivated by the difficulty of private image generation from only a relatively small private dataset (Xie et al., 2018; Jordon et al., 2018; Long et al., 2021; Chen et al., 2020).

Observe that the learning algorithm can use $D^{\mathrm{pub}}$ as it sees fit, and hence is easy if $D^{\mathrm{pub}}$ and $D^{\mathrm{priv}}$ are drawn from the same distribution. The challenge however is this may not be true, and often is not, in real-world settings. In this work, we relax this by assuming that *the support of the public data distribution contains the support of the private.* An example of this is when the public data comes from a general-purpose internet-scale source, while the private data consists of images of a specific type. For example, if the private dataset contains images of different objects, ImageNet (Deng et al., 2009) is a reasonable public dataset that includes a wide variety of objects.

## 3 METHOD

**Main ideas.** Our core insight is that instead of privately learning to generate highly detailed images from scratch, it is more sample-efficient to privately *adapt* a generative model trained on public data. This adaptation process can be broken into three steps (see Figure 1 for the illustration):

1. We use a pair of encoder and decoder trained entirely on public data. The encoder maps images to a low-dimensional feature space and the decoder generates images given a feature.
2. The encoder-decoder architecture effectively reduces the problem of learning the private image distribution to learning the private *feature* distribution in the encoder feature space. This can be done either by modeling the private features as a multivariate Gaussian, or by modeling the difference between the public and private feature distributions using a density ratio estimator. Both algorithms are highly sample-efficient when applied differentially privately.
3. Using the estimated private feature distribution, we can sample from it to obtain feature vectors and use the decoder to generate a novel image.

**Encoder-decoder architecture.** Given an image $\mathbf{x}$, the encoder is a neural network that maps $\mathbf{x}$ to some feature vector $\mathbf{v} = h(\mathbf{x})$ in a low-dimensional feature space. For instance, we can instantiate the encoder with the convolutional layers of a ResNet50 (He et al., 2016) network. The decoder is a generative model that takes the feature vector $\mathbf{v}$ and generates an image similar to $\mathbf{x}$, which can be done using an IC-GAN (Casanova et al., 2021). Both the encoder (*i.e.*, feature extractor) and the decoder are trained on public data using standard training methodologies.

The encoder-decoder architecture can be readily used for *non-private* image generation. In Figure 1(b), the private data is converted to a private feature distribution $\mathcal{P}^{\text{priv}}$ using the feature extractor $h$. To generate an image, we can sample $\mathbf{v} \sim \mathcal{P}^{\text{priv}}$ and use the IC-GAN $g$ to output an image. Notably, since IC-GAN is capable of generating *new images* whose feature vectors are close to $\mathbf{v}$, this allows us to obtain new samples from the private distribution.

**Private adaptation via density estimation.** The generation process outlined above is not differentially private since it depends on particular samples in the private data distribution. However, if we can estimate $\mathcal{P}^{\text{priv}}$ by modeling it privately then we can sample feature from the estimated distribution to generate new samples. Since $\mathcal{P}^{\text{priv}}$ is a distribution on a low-dimensional feature space, we can model it privately in a sample-efficient manner by leveraging existing DP techniques. In Sections 3.1 and 3.2, we propose two modeling algorithms: 1. *DP Multivariate Gaussian Estimation* (DP-MGE), which models $\mathcal{P}^{\text{priv}}$ as a multivariate Gaussian; and 2. *DP Density Ratio Estimation* (DP-DRE), which models the ratio between $\mathcal{P}^{\text{priv}}$ and the public feature distribution $\mathcal{P}^{\text{pub}}$ using a trained network. Figure 1(c) gives an illustration of the two algorithms.

**Private image generation.** Given a differentially private estimator of the private feature distribution $\mathcal{P}^{\text{priv}}$, we can sample from it to obtain a feature vector $\mathbf{v}$ and use the decoder to output a new generated image $g(\mathbf{v})$ just as in non-private generation. Figure 1(d) details the sampling procedure for both DP-MGE and DP-DRE. The only remaining question is how to model $\mathcal{P}^{\text{priv}}$ differentially privately, which we detail in the following sections.

## 3.1 DIFFERENTIALLY PRIVATE MULTIVARIATE GAUSSIAN ESTIMATION (DP-MGE)

Our first idea is to simply model $\mathcal{P}^{\text{priv}}$ as a normal distribution $\mathcal{N}(\boldsymbol{\mu}, \text{diag}(\mathbf{s}))$ where $\text{diag}(\mathbf{s})$ is a diagonal matrix with the diagonal $\mathbf{s}$. This is a plausible model for $\mathcal{P}^{\text{priv}}$ if it is unimodal, *e.g.*, if the private dataset $D^{\text{priv}}$ contains different breeds of dogs. Denote $\mathbf{v}_i^{\text{priv}}$ as the feature vector of the $i^{th}$ data in $D^{\text{priv}}$. From samples $\{\mathbf{v}_1^{\text{priv}}, \cdots, \mathbf{v}_n^{\text{priv}}\}$, the non-private estimators for $\boldsymbol{\mu}$ and $\mathbf{s}$ are: $\boldsymbol{\mu} = \frac{1}{n} \sum_{i=1}^{n} \mathbf{v}_i^{\text{priv}}$ and $\mathbf{s} = \frac{1}{n} \sum_{i=1}^{n} (\mathbf{v}_i^{\text{priv}})^2 - \boldsymbol{\mu}^2$, where $(\cdot)^2$ denotes elementary-wise squaring.

We employ the Gaussian mechanism to estimate $\boldsymbol{\mu}$ and $\mathbf{s}$ privately. Assume that $\|\mathbf{v}_i^{\text{priv}}\| \leq 1$ for $i = 1, \cdots, n$. The following estimators are $(\varepsilon, \delta)$-DP:

$$\boldsymbol{\mu}^{\text{dp}} = \frac{1}{n} \sum_{i=1}^{n} \mathbf{v}_i^{\text{priv}} + \mathcal{N}\left(\mathbf{0}, \frac{4\sigma_{\varepsilon/2, \delta/2}^2}{n^2} I\right), \mathbf{s}^{\text{dp}} = \frac{1}{n} \sum_{i=1}^{n} (\mathbf{v}_i^{\text{priv}})^2 - (\boldsymbol{\mu}^{\text{dp}})^2 + \mathcal{N}\left(\mathbf{0}, \frac{4\sigma_{\varepsilon/2, \delta/2}^2}{n^2} I\right),$$

$$(1)$$

where $\sigma_{\varepsilon, \delta} = \frac{\sqrt{2 \log 1/\delta + 2\varepsilon} + \sqrt{2 \log 1/\delta}}{2\varepsilon}$. We formally state this in the following Claim that this estimator satisfies DP and put the proof in the appendix.

**Claim 1.** *If $\|\mathbf{v}_i^{\text{priv}}\| \leq 1$ for $i = 1, \cdots n$, then the estimators for $(\boldsymbol{\mu}^{\text{dp}}, \mathbf{s}^{\text{dp}})$ defined in Equation 1 are $(\varepsilon, \delta)$-DP w.r.t. the private image dataset $D^{\text{priv}}$.*

---

**Algorithm 1** Differentially private training of density ratio estimator between public and private data.

1: **Input:** public feature vectors $\{\mathbf{v}_1^{\mathrm{pub}}, \cdots, \mathbf{v}_m^{\mathrm{pub}}\}$, private feature vectors $\{\mathbf{v}_1^{\mathrm{priv}}, \cdots, \mathbf{v}_n^{\mathrm{priv}}\}$, DP parameters $(\varepsilon, \delta)$
2: **Training hyperparameters:** total iteration $T$, learning rate $\eta$, batch size $B$, norm bound $C$.
3: Compute the proper $\sigma$ that guarantees the output to be $(\varepsilon, \delta)$-DP from Mironov et al. (2019) and Yousefpour et al. (2021).
4: Initialize the weight $\theta_0$, the $1^{\mathrm{st}}$ moment vector $m_0$, the $2^{\mathrm{nd}}$ moment vector $v_0$.
5: **for** $t = 1, \cdots, T$ **do**
6:    Sample a batch of private feature vectors $\{\mathbf{v}_{r_1}^{\mathrm{priv}}, \cdots \mathbf{v}_{r_{B_t}}^{\mathrm{priv}}\}$ with the poisson sample rate $q$.
7:    Uniformly sample $B_t$ public feature vectors $\{\mathbf{v}_{s_1}^{\mathrm{pub}}, \cdots \mathbf{v}_{s_{B_t}}^{\mathrm{pub}}\}$.
8:    Compute the gradient $g^t$ by $g^t \leftarrow \frac{1}{B_t} \left[ \sum_{i=1}^{B_t} \frac{g_i^t}{\max\{1, \|g_i^t\|/C\}} + \mathcal{N}\left(0, \sigma^2 C^2 \cdot I\right) \right]$, where $g_i^t = \nabla_\theta \left[ \log D_\theta(\mathbf{v}_{r_i}^{\mathrm{priv}}) + \log\left(1 - D_\theta(\mathbf{v}_{s_i}^{\mathrm{pub}})\right) \right]$.
9:    Update $m_t, v_t, \theta_t$ according to Adam.
10: **end for**
11: **Output:** $D_\theta^\dagger := \frac{D_\theta}{1 - D_\theta}$.

---

**Sampling.** With $\boldsymbol{\mu}^{\mathrm{dp}}$ and $\mathbf{s}^{\mathrm{dp}}$, we can generate a new latent vector $\mathbf{v}$ by sampling from $\mathcal{N}(\boldsymbol{\mu}^{\mathrm{dp}}, \mathrm{diag}(\mathbf{s}^{\mathrm{dp}}))$, and then use the IC-GAN to generate a new image $g(\mathbf{v})$.

### 3.2 DIFFERENTIALLY PRIVATE DENSITY RATIO ESTIMATION (DP-DRE)

The DP-MGE estimator proposed in Section 3.1 is simple and sample-efficient under a wide range of privacy budgets. However, when $\mathcal{P}_{\mathbf{v}}^{\mathrm{priv}}$ is more complicated and multi-modal, modeling it as a unimodal Gaussian distribution can suffer from a high bias.

We address this issue in DP-DRE, where we make additional use of the public data to model the private feature distribution. We use the encoder to map the public dataset $D^{\mathrm{pub}}$ to a feature distribution $\mathcal{P}^{\mathrm{pub}}$ and then model the difference between $\mathcal{P}^{\mathrm{pub}}$ and $\mathcal{P}^{\mathrm{priv}}$. For instance, if $D^{\mathrm{pub}}$ is the ImageNet dataset and $D^{\mathrm{priv}}$ contains only birds, we only need to train a *discriminator* to filter out all non-bird samples from ImageNet and use the features of the remaining samples to generate new images.

**Density ratio estimation.** To model the difference between the public and private data distributions, we propose estimating the ratio $\mathcal{P}^{\mathrm{priv}}(\mathbf{v}) / \mathcal{P}^{\mathrm{pub}}(\mathbf{v})$ by training a discriminator (similar to those used in GAN training) $D_\theta$ to minimize:

$$\min_\theta \frac{1}{n} \sum_{i=1}^n \left[ \log D_\theta(\mathbf{v}_i^{\mathrm{priv}}) \right] + \frac{1}{m} \sum_{j=1}^m \left[ \log \left(1 - D_\theta(\mathbf{v}_j^{\mathrm{pub}})\right) \right], \quad (2)$$

where $\mathbf{v}_i^{\mathrm{priv}} = h(\mathbf{x}_i^{\mathrm{priv}})$ and $\mathbf{v}_j^{\mathrm{pub}} = h(\mathbf{x}_j^{\mathrm{pub}})$ are features for the $i^{th}$ private data and $j^{th}$ public data. The loss function for the discriminator $D_\theta$ in Equation 2 is the empirical loss for the objective:

$$\mathbb{E}_{\mathbf{v} \sim \mathcal{P}^{\mathrm{priv}}} \left[ \log D_\theta(\mathbf{v}) \right] + \mathbb{E}_{\mathbf{v} \sim \mathcal{P}^{\mathrm{pub}}} \left[ \log \left(1 - D_\theta(\mathbf{v})\right) \right].$$

Importantly, Goodfellow et al. (2014) showed that under this objective, the optimal discriminator is $D^*(\mathbf{v}) = \frac{\mathcal{P}^{\mathrm{priv}}(\mathbf{v})}{\mathcal{P}^{\mathrm{priv}}(\mathbf{v}) + \mathcal{P}^{\mathrm{pub}}(\mathbf{v})}$. Hence using a trained discriminator with $D_\theta \approx D^*$, we can define the density ratio estimator $D_\theta^\dagger(\mathbf{v}) := \frac{D_\theta(\mathbf{v})}{1 - D_\theta(\mathbf{v})}$, with $D_\theta^\dagger(\mathbf{v}) = \frac{D_\theta(\mathbf{v})}{1 - D_\theta(\mathbf{v})} \approx \frac{D^*(\mathbf{v})}{1 - D^*(\mathbf{v})} = \frac{\mathcal{P}^{\mathrm{priv}}(\mathbf{v})}{\mathcal{P}^{\mathrm{pub}}(\mathbf{v})}$ which recovers the desired density ratio.

**Sampling.** The density ratio estimator allows us to approximate $\mathcal{P}^{\mathrm{priv}}(\mathbf{v}) \approx D_\theta^\dagger(\mathbf{v}) \mathcal{P}^{\mathrm{pub}}(\mathbf{v})$, which is a re-weighting of the public feature distribution $\mathcal{P}^{\mathrm{pub}}$. Thus, to sample approximately from $\mathcal{P}^{\mathrm{priv}}$, we can define an empirical distribution over $\{\mathbf{v}_1^{\mathrm{pub}}, \ldots, \mathbf{v}_m^{\mathrm{pub}}\}$ with sampling probability $\mathbf{p} = \frac{1}{N}(D_\theta^\dagger(\mathbf{v}_1^{\mathrm{pub}}), \ldots, D_\theta^\dagger(\mathbf{v}_m^{\mathrm{pub}}))$, where $N$ is a normalization factor. A new image from the private distribution can then be generated using the IC-GAN decoder.

**Private discriminator training.** To make the density ratio estimator training private, we employ DP-SGD (Abadi et al., 2016) with the Adam optimizer (Kingma & Ba, 2014). We use the pri-

vacy accounting method in Mironov et al. (2019) and Balle et al. (2020) and its implementation in Yousefpour et al. (2021) for DP-SGD. Algorithm 1 gives a summarization of the training algorithm.

# 4 EXPERIMENT

We evaluate our methods and demonstrate their practical applicability in realistic image generation[1]. Our evaluation aims to answer the following questions: **1.** How do our methods compare with existing baselines in terms of standard image-generation metrics and image quality? **2.** How does DP-MGE compare with DP-DRE? **3.** How do DP-DRE and DP-MGE perform when the assumption about the distribution supports does not hold?

## 4.1 EXPERIMENT SET-UP

**Datasets.** For all our experiments, we use ImageNet (Deng et al., 2009) as the public data, with the pretrained ResNet50 feature extractor $h$ and IC-GAN $g$ from Casanova et al. (2021)[2]. We use multiple private datasets – CIFAR10 (Cifar10; (Krizhevsky et al., 2009)), Oxford-IIIT Pet Dataset (Pet; (Parkhi et al., 2012)), Stanford Cars Dataset (Car; (Krause et al., 2013)), Caltech-UCSD Birds Dataset (Bird; (Wah et al., 2011)), Nico+ (Zhang et al., 2022) with Grass (Objects-grass) and Nico+ with Autumn (Objects-autumn). These are considerably smaller than Imagenet, and have sizes 50000, 3680, 5992, 8144, 16256 and 7272 respectively. Images in Cifar10 have resolution 32 and we resize images from the other datasets to resolution 128. We use the train split of each dataset for training the image generation algorithms and validation or test splits for evaluation.

**Algorithm set-up.** We evaluate both DP-MGE and DP-DRE. For DP-MGE, the normalization operator inside the feature extractor $h$, as implemented in the IC-GAN, ensures that the norm of the features $\|\mathbf{v}_i\| \leq 1$, thus ensuring privacy (see Claim 1). For DP-DRE, we choose a two-layer MLP as our discriminator $D_\theta^\dagger$. More training details are provided in the appendix.

**Baselines.** We compare our methods with the following three baselines that all utilize public data; details on how they are trained are provided in the appendix. In all baselines, for fair comparison, we make our best to reuse IC-GAN as much as possible, which is aimed to demonstrate the efficiency of our method in terms of leveraging both the public data and IC-GAN. We also report a non-private baseline: the scores for images generated from the IC-GAN when the private data is directly input to it. This is an upper bound for any IC-GAN based image generation algorithm.

*1. DP finetuning on private data* (DP-GAN-FT) finetunes a pretrained public GAN on private data with differential privacy. For our experiment, we train a GAN on ImageNet in the feature space described by the pretrained feature extractor $h$. Both the generator and discriminator of this GAN are 4-layer MLPs. We sequentially combine the generator of this trained GAN with the IC-GAN generator, and its discriminator with the IC-GAN discriminator to get a complete unconditional GAN on ImageNet. The combined generator and discriminator are then finetuned together on the private data using a differentially private version of Adam.

*2. DP mean embedding with perceptual features* (DP-MEPF; (Harder et al., 2022)) uses the public data to extract pretrained features, and calculates the first and second moments of the private data in the feature space with differential privacy. A generative model is then trained to generate data that matches these moments. For CIFAR10, we follow the same set-up as in Harder et al. (2022) – the pre-trained features are the perceptual features from each layer of the VGG-19 network (Simonyan & Zisserman, 2014) and the generator is a ResNet. For the other datasets, we pretrain a ResNet50 on ImageNet. We use BigGAN as the architecture of the generator and set the deep features as the output of the layer right before the last pooling layer.

*3. DP-GAN with model inversion* (DP-GAN-MI; (Chen et al., 2021)) pretrains a GAN on the public data and then trains a differentially private GAN in its latent space. To generate a new image, it first generates a latent vector via the private GAN, and passes it to the pretrained GAN. We use this

---

[1]Our code is released at https://anonymous.4open.science/r/dp_img_gen_with_public_data-8FC5/README.md

[2]The pretrained ResNet50 feature extractor is from https://github.com/facebookresearch/ic_gan, and we train the IC-GAN on face-blurred ImageNet using code from the same repo.

Table 1: FID score (lower is better) of generated data for 6 different private datasets.

| Method | Cifar10 | | | | | Pet | | | | |
|---|---|---|---|---|---|---|---|---|---|---|
| | $\varepsilon = \infty$ | $\varepsilon = 10$ | $\varepsilon = 3$ | $\varepsilon = 1$ | $\varepsilon = 0.1$ | $\varepsilon = \infty$ | $\varepsilon = 10$ | $\varepsilon = 3$ | $\varepsilon = 1$ | $\varepsilon = 0.1$ |
| Non-private IC-GAN | 16.6 | | | | | 29.7 | | | | |
| DP-GAN-FT | 33.8 | 36.7 | 36.7 | 38.3 | 38.3 | 46.8 | 103.3 | 103.0 | 104.7 | 103.6 |
| DP-MEPF | 36.9 | 37.0 | 41.8 | 68.3 | 296.5 | 95.2 | 89.3 | 91.4 | 104.2 | 217.9 |
| DP-GAN-MI | **19.5** | 97.2 | 94.4 | 75.6 | 150.7 | **29.1** | 127.2 | 189.6 | 209.7 | 192.2 |
| DP-MGE | 33.9 | 43.3 | 39.9 | 39.9 | 51.1 | 77.1 | 74.8 | 76.1 | 109.7 | 166.4 |
| DP-DRE | 19.8 | **21.2** | **21.0** | **20.9** | **20.9** | 30.7 | **33.0** | **34.3** | **32.7** | **93.7** |

| Method | Car | | | | | Bird | | | | |
|---|---|---|---|---|---|---|---|---|---|---|
| | $\varepsilon = \infty$ | $\varepsilon = 10$ | $\varepsilon = 3$ | $\varepsilon = 1$ | $\varepsilon = 0.1$ | $\varepsilon = \infty$ | $\varepsilon = 10$ | $\varepsilon = 3$ | $\varepsilon = 1$ | $\varepsilon = 0.1$ |
| Non-private IC-GAN | 17.7 | | | | | 20.5 | | | | |
| DP-GAN-FT | 26.5 | 150.5 | 148.6 | 149.3 | 149.3 | 20.4 | 109.8 | 109.9 | 109.7 | 109.7 |
| DP-MEPF | 58.4 | 52.4 | 44.5 | 52.4 | 140.5 | 52.9 | 65.3 | 65.0 | 67.1 | 97.2 |
| DP-GAN-MI | **15.8** | 38.8 | 62.6 | 239.9 | 239.1 | **19.4** | 71.1 | 78.0 | 206.2 | 190.4 |
| DP-MGE | 20.0 | 18.8 | 19.6 | 43.1 | 203.3 | 27.1 | 26.3 | 27.1 | 64.7 | 166.5 |
| DP-DRE | 17.8 | **18.6** | **18.4** | **17.3** | **68.3** | 20.7 | **21.6** | **23.0** | **24.3** | **86.6** |

| Method | Objects-Grass | | | | | Objects-Autumn | | | | |
|---|---|---|---|---|---|---|---|---|---|---|
| | $\varepsilon = \infty$ | $\varepsilon = 10$ | $\varepsilon = 3$ | $\varepsilon = 1$ | $\varepsilon = 0.1$ | $\varepsilon = \infty$ | $\varepsilon = 10$ | $\varepsilon = 3$ | $\varepsilon = 1$ | $\varepsilon = 0.1$ |
| Non-private IC-GAN | 20.8 | | | | | 38.4 | | | | |
| DP-GAN-FT | 31.6 | 57.8 | 58.8 | 56.9 | 57.9 | 44.1 | 70.2 | 69.8 | 71.3 | 69.9 |
| DP-MEPF | 92.3 | 80.7 | 76.7 | 84.7 | 101.2 | 85.5 | 82.7 | 95.0 | 126.6 | 159.9 |
| DP-GAN-MI | **24.2** | 86.3 | 79.0 | 89.2 | 145.1 | **40.6** | 80.7 | 125.2 | 124.6 | 157.7 |
| DP-MGE | 56.3 | 51.0 | 50.8 | 50.1 | 87.2 | 74.2 | 71.1 | 69.5 | 73.2 | 113.5 |
| DP-DRE | 25.5 | **26.3** | **26.7** | **27.4** | **28.9** | 44.9 | **46.7** | **46.7** | **48.3** | **53.7** |

Table 2: Precision and recall (higher is better) of our methods and baselines on Cifar10 dataset.

| Method | $\varepsilon = \infty$ | | $\varepsilon = 10$ | | $\varepsilon = 3$ | | $\varepsilon = 1$ | | $\varepsilon = 0.1$ | |
|---|---|---|---|---|---|---|---|---|---|---|
| | Prec. | Rec. | Prec. | Rec. | Prec. | Rec. | Prec. | Rec. | Prec. | Rec. |
| Non-private IC-GAN | Precision: 0.971, Recall: 0.969 | | | | | | | | | |
| DP-GAN-FT | 0.854 | 0.889 | 0.880 | 0.884 | 0.867 | 0.870 | 0.876 | 0.890 | 0.876 | 0.875 |
| DP-MEPF | 0.929 | 0.879 | 0.922 | 0.880 | 0.908 | 0.847 | 0.825 | 0.638 | 0.011 | 0.000 |
| DP-GAN-MI | **0.965** | **0.961** | 0.521 | 0.544 | 0.540 | 0.420 | 0.785 | 0.541 | 0.217 | 0.444 |
| DP-MGE | 0.886 | 0.697 | 0.885 | 0.719 | 0.908 | 0.727 | 0.898 | 0.739 | 0.836 | 0.733 |
| DP-DRE | 0.950 | 0.933 | **0.946** | **0.925** | **0.943** | **0.926** | **0.944** | **0.933** | **0.946** | **0.936** |

procedure with the IC-GAN architecture, and train a differentially private GAN in the latent space of the feature extractor $h$. The generator and discriminator both are 4-layer MLPs.

**Evaluation metrics.** Since there is no one perfect metric for evaluating generation quality, we use three popular metrics to measure the effectiveness of our methods: *Frechet Inception Distance* (FID; lower is better) score (Heusel et al., 2017), *Precision and Recall* (higher is better) (Sajjadi et al., 2018), and *Number of Different Bins* (NDB; lower is better)(Richardson & Weiss, 2018). The details of these metrics are in the appendix.

## 4.2 RESULTS

We run methods on each dataset with $\varepsilon \in \{\infty, 10, 3, 1, 0.1\}$ and $\delta = 10^{-5}$. Table 1 reports the FID scores. Table 2 and Figure 3 present the precision and recall and NDB on Cifar10; the results for the other five datasets are presented in the appendix. We find that the conclusions drawn from the precision and recall and NDB tables largely agree with the results of the FID scores.

**Quantitative comparison with baselines.** Without privacy guarantee ($\varepsilon = \infty$), we see that DP-GAN-MI outperforms all other methods at all datasets, which shows that in the absence of privacy, GAN has the strongest ability to learn an arbitrary distribution. The FID scores of DP-DRE are close to DP-GAN-MI on all datasets; this suggests the bias of DP-DRE is almost as small as that of a strong generative method, e.g. GAN. In contrast, DP-MGE suffers from a larger bias as expected.

However, at higher privacy levels ($\varepsilon \leq 10$), our methods DP-MGE and DP-DRE fare better than all three baselines. DP-MEPF already has a large bias at $\varepsilon = \infty$ for all datasets except Cifar10. This might be because it is difficult to adapt to high resolution images. DP-GAN-FT and DP-GAN-MI

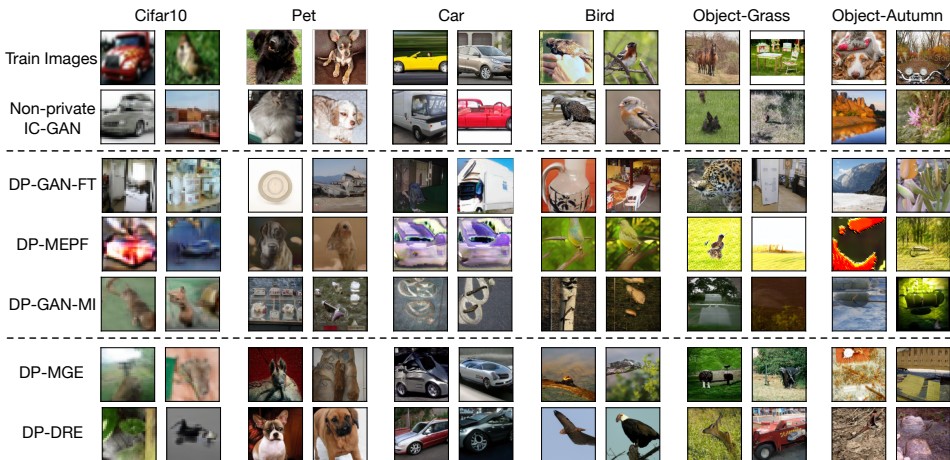

Figure 2: Examples of all algorithms ($\varepsilon = 1$) across six datasets.

also performs poorly with DP noise. The training process of GANs is known to be quite brittle, and so this might be because DP makes the training even harder. Moreover, the goal of DP-GAN-FT and DP-GAN-MI is hard: they are to learn the entire private distribution. In contrast, DP-MGE and DP-DRE have more stable training processes and simpler goals to learn: DP-MGE is to learn the first and second moments of the distribution in the feature space and DP-DRE is to learn the difference between the public and the private distributions.

**Image quality comparison with baselines.** Figure 2 shows the randomly generated examples for each algorithm with $\varepsilon = 1$. More examples with different $\varepsilon$ are in the appendix. We see that DP-DRE and DP-MGE generate the most in-distribution images with high quality for all datasets. Other methods either generate many irrelevant out-of-distribution images (DP-GAN-FT, DP-GAN-MI) or have many artifacts (DP-MEPF). These artifacts may be due to the fact that DP-MEPF does not use a pretrained public encoder.

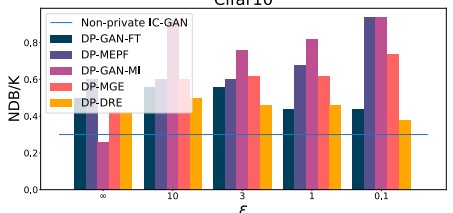

Figure 3: The percentage of different bins NDB/K (lower is better) of our methods and baselines on Cifar10 dataset.

**Comparison between DP-MGE and DP-DRE.** Because gaussian distribution might have pool approximation to complicated distribution, we expect DP-MGE to have a larger bias than DP-DRE. This is what we observe. DP-DRE is considerably better than DP-MGE on the datasets with multiple different objects (Cifar10, Pet, Objects-Grass, Objects-Autumn). Moreover, we can observe that for some datasets, DP-DRE has better robustness at small $\varepsilon$ than DP-MGE. This looks counterintuitive because DP-MGE has less parameters to estimate. Nevertheless, DP-DRE by its design is indeed more robust to the noise perturbation: any noise perturbation for the mean and variance estimators in DP-MGE will be reflected in the distribution to be generated, while DP-DRE essentially learns a boundary to select representative data from public data and when the representative data has a certain margin with the remaining data, a noise perturbation on the boundary may not influence the data selection.

**Sanity check for DP-DRE.** To check whether DP-DRE learns meaningful weights $\mathbf{p}$, we compare DP-DRE with a naive baseline: uniformly sample feature vectors from ImageNet validation set and get images from IC-GAN with these feature vectors. The FID scores of this baseline are 37.9 (Cifar10), 100.1 (Pet), 147.5

Table 3: The weight assigned to same semantic superclass.

|  | $\varepsilon = \infty$ | $\varepsilon = 10$ | $\varepsilon = 3$ | $\varepsilon = 1$ | $\varepsilon = 0.1$ |
|---|---|---|---|---|---|
| Bird | 0.98 | 0.99 | 0.98 | 0.97 | 0.18 |
| Car | 0.97 | 0.96 | 0.96 | 0.98 | 0.58 |

(Car), 111.3 (Bird), 49.0 (Objects-grass) and 63.5 (Objects-autumn). By comparing them with Table 1, even with $\varepsilon = 1.0$, DP-DRE achieves much better FID score on all datasets. This means that DP-DRE learns meaningful weights to approximate the private distribution.

Moreover, to see to what extent DP-DRE generates in-distribution images, we record the weights assigned by DP-DRE to images in the ImageNet validation set. For the Bird dataset, we sum up

the weights for images belonging to the superclass *bird*. For the Car dataset, we similarly sum up the weights belonging to the superclass *wheeled vehicle*. The results in Table 3 indicate that at least $96\%$ of the images generated are close to that of Birds or Cars when the $\varepsilon \geq 1.0$.

**Rejection sampling in the latent space versus the image space.** In fact, DP-DRE can be viewed as a form of (efficient) rejection sampling in the latent space of the IC-GAN. We also made a first attempt to learn an image space discriminator (WideResNet16-4) between public IC-GAN and private images and apply rejection sampling on public IC-GAN. The FID scores are not good even when $\varepsilon = \infty$: 38.3 (Car), 120.6 (Pet) and 119.4 (Bird). Indeed, this is expected for two reasons. First, the rejection sampler requires accurate probability scores, while probability scores from deep neural networks (e.g. WideResNet16-4 has 2.7M parameters) need careful calibration (Guo et al., 2017); Second, the deep neural network has overfitting problems. We have checked that on the pet and bird dataset, the model only learns to assign high probability scores to private images, but not to images generated from IC-GAN in these categories. Instead, the discriminator in DP-DRE is only an MLP with a width no larger than 16, so it does not encounter these two obstacles. Besides the image generation performance, sampling computational efficiency becomes to an issue as well. Rejection sampling in the image space needs a lot of samples from IC-GAN. For example, to get one sample on the Car dataset, it requires $\sim 1400$ samples generated from IC-GAN on average. By contrast, to get a single sample, DP-DRE only requires one sample from IC-GAN.

**Image generation results when the assumption is severely violated.** The success of DP-DRE relies on the assumption: the support of the private distribution is contained by the support of the public distribution. DP-DRE has great performance on the private datasets so far, because these datasets are closed to some subsets of ImageNet. We further considered two datasets very different from ImageNet: Chest X-Ray Images (Pneumonia) (Kermany et al., 2018) and Describable Textures Dataset (Cimpoi et al., 2014). As expected, even the FID scores of non-private IC-GAN, which serves as the performance upper bound for DP-DRE, DP-MGE and DP-GAN-MI, are very high: 221.4 and 68.7. This shows the necessity of the assumption for DP-MGE and DP-DRE.

## 5 Related Work

**DP image generation.** There has been a body of work on differentially private GAN-training solely from private data (Xie et al., 2018; Jordon et al., 2018; Chen et al., 2020; Long et al., 2021; Wang et al., 2021). Cao et al. (2021) and Harder et al. (2021) explore DP-generation algorithms that use an alternative loss or match the first and second feature moments. Ghalebikesabi et al. (2023) and Dockhorn et al. (2022) study the diffusion model with DP. Unfortunately, these work are far away from generating realistic high resolution images with high quality with the reliable DP guarantees.

**DP classifier training with the public data.** A line of work has looked into this setting in order to balance a privacy vs. classification accuracy tradeoff. Examples that public data and private data have the same distribution include the PATE framework (Papernot et al., 2016; 2018) as well as its extensions (Zhu et al., 2020). Tramer & Boneh (2020) studies the usage of public data which have different distribution from private data.

**Transfer learning in GAN.** GAN transfer has been investigated to train a generative model with the limited data. Wang et al. (2018; 2020); Zhao et al. (2020) and Mo et al. (2020) propose different finetuning strategies to transfer knowledge from a pretrained unconditional GAN. Shahbazi et al. (2021); Laria et al. (2022) and Dinh et al. (2022) study the transfer between the conditional GAN. None of these work involve differential privacy.

## 6 Conclusion

This paper studies how to use *generic* large-scale public data to improve the DP image generation. Our methods apply under the realistic assumption that the support of the public data contains the one of the private. Our empirical evaluations show our methods achieve SOTA for DP image generation.

**Limitations and future work.** Methods proposed in this work rely on the performance of pretrained IC-GAN and the assumption of public and private support. Thus, one potential direction is to extend our work to more sophisticated generative methods such as diffusion models. Another direction is to relax this assumption to wider varieties of public data, which may have more use-cases.

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

## A  PROOFS IN SECTION 3

**Claim 2.** *If $\|\mathbf{v}_i^{\mathrm{priv}}\| \leq 1$ for $i = 1, \cdots n$, then the estimators for $(\mu^{\mathrm{dp}}, \mathbf{s}^{\mathrm{dp}})$ defined in Equation (1) and (2) are $(\varepsilon, \delta)$-DP w.r.t. the private image dataset $D^{\mathrm{priv}}$.*

*Proof of Claim 1.* We will first show that the gaussian machanism $f(D) + \mathcal{N}(0, S^2 \sigma_{\varepsilon,\delta}^2 I)$ with $\sigma_{\varepsilon,\delta} = \frac{\sqrt{2\log 1/\delta + 2\varepsilon} + \sqrt{2\log 1/\delta}}{2\varepsilon}$ guarantees $(\varepsilon, \delta)$-DP for any $\varepsilon > 0$. Notice that this is different from the commonly used Gaussian mechanism where $\sigma_{\varepsilon,\delta} = \frac{\sqrt{2\log 1.25/\delta}}{\varepsilon}$ (Dwork et al., 2014), which only holds for $\varepsilon < 1$. According to Renyi-DP (RDP) paper, Gaussian mechanism guarantees $(\alpha, \alpha/(2\sigma^2))$-RDP, which is equivalent to $\left(\alpha/(2\sigma^2) + \frac{\log 1/\delta}{\alpha - 1}, \delta\right)$-DP. $\alpha/(2\sigma^2) + \frac{\log 1/\delta}{\alpha - 1}$ achieves the minimum $\sqrt{\frac{2\log 1/\delta}{\sigma^2}} + 1/(2\sigma^2)$ at $\alpha = 1 + \sqrt{2\sigma^2 \log 1/\delta}$. Thus the guassian mechanism guarantees $\left(\sqrt{\frac{2\log 1/\delta}{\sigma^2}} + 1/(2\sigma^2), \delta\right)$-DP. In the other word, $\sigma_{\varepsilon,\delta} = \frac{\sqrt{2\log 1/\delta + 2\varepsilon} + \sqrt{2\log 1/\delta}}{2\varepsilon}$ guarantees $(\varepsilon, \delta)$-DP.

With the above result, the only remaining thing is to derive the sensitivity of the $\frac{1}{n}\sum_{i=1}^{n} \mathbf{v}_i^{\mathrm{priv}}$ and $\frac{1}{n}\sum_{i=1}^{n} \mathbf{v}_i^{\mathrm{priv}}$. Suppose the only difference between two neibouring datasets is $\left(\mathbf{v}_i^{\mathrm{priv}}, \left(\mathbf{v}_i^{\mathrm{priv}}\right)'\right)$. The sensitivity of $\frac{1}{n}\sum_{i=1}^{n} \mathbf{v}_i^{\mathrm{priv}}$ is

$$\left\|\frac{1}{n}\sum_{i=1}^{n} \mathbf{v}_i^{\mathrm{priv}} - \frac{1}{n}\sum_{i=1}^{n} \left(\mathbf{v}_i^{\mathrm{priv}}\right)'\right\| = \frac{1}{n}\left\|\mathbf{v}_i^{\mathrm{priv}} - \left(\mathbf{v}_i^{\mathrm{priv}}\right)'\right\| \leq \frac{1}{n}\left\|\mathbf{v}_i^{\mathrm{priv}}\right\| + \frac{1}{n}\left\|\left(\mathbf{v}_i^{\mathrm{priv}}\right)'\right\| \leq \frac{2}{n}.$$

The sensitivity of $\frac{1}{n}\sum_{i=1}^{n} (\mathbf{v}_i^{\mathrm{priv}})^2$ is

$$\left\|\frac{1}{n}\sum_{i=1}^{n} (\mathbf{v}_i^{\mathrm{priv}})^2 - \frac{1}{n}\sum_{i=1}^{n} \left(\left(\mathbf{v}_i^{\mathrm{priv}}\right)'\right)^2\right\| \leq \frac{1}{n}\left\|(\mathbf{v}_i^{\mathrm{priv}})^2\right\| + \frac{1}{n}\left\|\left(\left(\mathbf{v}_i^{\mathrm{priv}}\right)'\right)^2\right\| \leq \frac{2}{n}$$

Then $\mu^{\mathrm{dp}}$ and $\mathbf{s}^{\mathrm{dp}}$ are $(\varepsilon/2, \delta/2)$-DP and they together are $(\varepsilon, \delta)$-DP

$\square$

## B  DETAILS OF METRICS NDB

For completeness, we revisit the details of evaluation metric.

The *Frechet Inception Distance* (FID; (Heusel et al., 2017)) Score measures the distance between the generated distribution and the target distribution by comparing the means and variances in a feature space computed from a pretrained Inception V3 model.

The *Precision and Recall* (Sajjadi et al., 2018) metric evaluates the distance between the target and the generated distributions along two separate dimensions – precision, which measures the sample quality of the generation algorithm, and recall, which measures the proportion of the target distribution covered by the generated distribution. These two are evaluated in a feature space after clustering; as suggested by Sajjadi et al. (2018), we choose $K = 20$ clusters and report the maximum of $F_8$ and $F_{1/8}$ for precision and recall.

The *Number of Different Bins* (NDB; (Richardson & Weiss, 2018)) compares the histogram of two distributions in the pixel space after clustering samples from the target distribution into bins. The detailed steps are:

---

**Algorithm 2** Differentially private training of GAN.

1: **Input:** Data points $\{\mathbf{x}_1, \cdots, \mathbf{x}_n\}$, DP parameters $(\varepsilon, \delta)$
2: **Training hyperparameters:** total training iteration $T$, learning rate $\eta$, batch size $B$, norm bound $C$.
3: Initialize the weight $\theta = (\theta^{\text{gen}}, \theta^{\text{dis}})$, the 1$^{\text{st}}$ moment vector $m = (m^{\text{gen}}, m^{\text{dis}})$, the 2$^{\text{nd}}$ moment vector $v = (v^{\text{gen}}, v^{\text{dis}})$.
4: Compute the proper $\sigma$ that guarantees the output to be $(\varepsilon, \delta)$-DP from Mironov et al. (2019); Yousefpour et al. (2021).
5: **for** $t = 1, \cdots, T$ **do**
6:     **for** $\tau = 1, \cdots 5$ **do**
7:         Sample a batch of real data points $\{\mathbf{x}_{r_1}, \cdots \mathbf{x}_{r_{B_t}}\}$ with the poisson sample rate $B/n$
8:         Uniformly sample $B_t$ fake data points $\{\mathbf{x}'_1, \cdots \mathbf{x}'_{B_t}\}$, where $\mathbf{x}'_i = G_{\theta^{\text{gen}}}(\mathbf{z}_i)$ is sampled from current generative model $G_{\theta^{\text{gen}}}$.
9:         Compute the gradient

$$g^t \leftarrow \frac{1}{B_t} \left[ \sum_{i=1}^{B_t} \frac{g_i^t}{\max\{1, \|g_i^t\|/C\}} + \mathcal{N}\left(0, \sigma^2 C^2 \cdot I\right) \right],$$

        where $g_i^t = \nabla_{\theta^{\text{dis}}} \left[ D_{\theta^{\text{dis}}}(\mathbf{x}_{r_i}) - D_{\theta^{\text{dis}}}(\mathbf{x}'_i) + \| \frac{\partial}{\partial \mathbf{x}'_{r_i}} D_{\theta^{\text{dis}}}(\mathbf{x}_{r_i}) \| \right]$.

10:         Update $m^{\text{dis}}, v^{\text{dis}}, \theta^{\text{dis}}$ according to Adam.
11:     **end for**
12:     Uniform sample B fake data points $G_{\theta^{\text{gen}}}(\mathbf{z}_1), \cdots, G_{\theta^{\text{gen}}}(\mathbf{z}_B)$
13:     $g^t \leftarrow \frac{1}{B} \sum_{i=1}^{B} \nabla_{\theta^{\text{gen}}} \left[ D_{\theta^{\text{dis}}}(G_{\theta^{\text{gen}}}(\mathbf{z}_i)) \right]$
14:     Update $m^{\text{gen}}, v^{\text{gen}}, \theta^{\text{gen}}$ according to Adam.
15: **end for**
16: **Output:** $G_{\theta^{\text{gen}}}$.

---

1. Cluster samples from the target distribution in the pixel space.

2. For each cluster $k$, we calculate the proportion $p_k$ of target samples that are assigned to this cluster

3. Assign the generated samples to the closest clusters and compute the proportion $p'_k$ similarly.

4. Measure the number of clusters that $p_k$ and $p'_k$ are significantly different.

When the learned distribution is closer to the target distribution, the clusters that $p_k$ and $p'_k$ are significantly different are less. We choose $K = 50$ clusters for the evaluation in our experiment.

## C    IMPLEMENTATION DETAILS OF METHOD AND BASELINES

We introduce the implementation details of different methods in the next paragraphs. In addition, DP-GAN-FT and DP-GAN-MI follow the same training procedures shown in Algorithm 2, except that DP-GAN-FT is initialized with a pretrained GAN while DP-GAN-MI is randomly initialized.

**DP-DRE:** The size of the validation set $V$ used in DP-DRE is 50000, 50 images per class in the ImageNet. The hidden widths $w$ of MLP are selected from $\{1, 4, 16\}$. As for the training hyperparameters in Algorithm 1 (main paper), we select the total training iteration $T \in \{3 \times 10^3, 10^4, 3 \times 10^4\}$, the learning rate $\eta \in 10^{-3}, 10^{-4}$ the batch size $B \in \{64, 256\}$, and the norm bound $C$ as 1.0.

**DP-GAN-FT:** The generator $g^{\text{latent}}$ in the GAN of the public latent distribution has the architecture of 4-layer MLP with the hidden width 1024 and latent dimensionality 128. The paired discriminator $D^{\text{latent}}$ is also a 4-layer MLP with the hidden width 1024. The GAN of public latent distribution is optimized by Adam with the learning rate of $10^{-4}$. As for the finetuning, we select batch size $B \in \{4, 16, 64\}$ and set the bounded norm $C$ as 1.0. The learning rate $\eta$ is selected from $\{10^{-5}, 10^{-6}, 10^{-7}\}$. The total number of training iterations are selected from $\{3 \times 10^3, 3 \times 10^4\}$. We evaluate the result every 30 epochs and save the *best* checkpoint along each training.

| Dataset | $\varepsilon$ | DP-DRE | | | | DP-GAN-FT | | | DP-MEPF | DP-GAN-MI | | | |
|---|---|---|---|---|---|---|---|---|---|---|---|---|---|
| | | $w$ | $T$ | $B$ | $\eta$ | $\eta$ | $T$ | $B$ | $\eta$ | $w$ | $T$ | $B$ | $\eta$ |
| **Cifar10** | $\infty$ | 16 | 3000 | 64 | 0.0001 | $10^{-7}$ | 30000 | 4 | $10^{-5}$ | 128 | 100000 | 64 | 0.001 |
| | 10 | 1 | 3000 | 64 | 0.001 | $10^{-6}$ | 3000 | 16 | $10^{-5}$ | 128 | 100000 | 64 | 0.0001 |
| | 3 | 1 | 3000 | 64 | 0.001 | $10^{-6}$ | 30000 | 4 | $10^{-5}$ | 32 | 10000 | 64 | 0.001 |
| | 1 | 1 | 3000 | 64 | 0.001 | $10^{-6}$ | 3000 | 4 | $10^{-5}$ | 32 | 3000 | 64 | 0.001 |
| | 0.1 | 16 | 3000 | 64 | 0.001 | $10^{-6}$ | 30000 | 4 | $10^{-5}$ | 128 | 10000 | 64 | 0.001 |
| **Pet** | $\infty$ | 4 | 10000 | 256 | 0.0001 | $10^{-6}$ | 30000 | 16 | $10^{-6}$ | 128 | 100000 | 64 | 0.0001 |
| | 10 | 16 | 3000 | 64 | 0.001 | $10^{-7}$ | 30000 | 4 | $10^{-5}$ | 32 | 3000 | 64 | 0.001 |
| | 3 | 16 | 3000 | 64 | 0.001 | $10^{-7}$ | 30000 | 4 | $10^{-6}$ | 128 | 3000 | 4 | 0.001 |
| | 1 | 4 | 10000 | 64 | 0.001 | $10^{-6}$ | 30000 | 4 | $10^{-6}$ | 128 | 10000 | 16 | 0.001 |
| | 0.1 | 16 | 10000 | 64 | 0.001 | $10^{-7}$ | 30000 | 4 | $10^{-6}$ | 128 | 10000 | 16 | 0.001 |
| **Car** | $\infty$ | 1 | 10000 | 64 | 0.001 | $10^{-6}$ | 30000 | 16 | $10^{-6}$ | 128 | 100000 | 4 | 0.0001 |
| | 10 | 1 | 30000 | 256 | 0.001 | $10^{-7}$ | 3000 | 16 | $10^{-5}$ | 32 | 10000 | 64 | 0.001 |
| | 3 | 4 | 10000 | 64 | 0.001 | $10^{-7}$ | 30000 | 4 | $10^{-6}$ | 32 | 3000 | 64 | 0.001 |
| | 1 | 4 | 30000 | 64 | 0.001 | $10^{-6}$ | 30000 | 16 | $10^{-6}$ | 128 | 10000 | 64 | 0.001 |
| | 0.1 | 16 | 10000 | 64 | 0.001 | $10^{-6}$ | 30000 | 4 | $10^{-6}$ | 128 | 3000 | 16 | 0.001 |
| **Bird** | $\infty$ | 1 | 10000 | 64 | 0.001 | $10^{-6}$ | 30000 | 4 | $10^{-6}$ | 32 | 100000 | 64 | 0.0001 |
| | 10 | 1 | 30000 | 64 | 0.001 | $10^{-5}$ | 3000 | 16 | $10^{-5}$ | 32 | 10000 | 64 | 0.001 |
| | 3 | 1 | 30000 | 64 | 0.001 | $10^{-7}$ | 30000 | 4 | $10^{-5}$ | 32 | 3000 | 64 | 0.001 |
| | 1 | 1 | 30000 | 64 | 0.001 | $10^{-7}$ | 30000 | 4 | $10^{-5}$ | 32 | 3000 | 16 | 0.001 |
| | 0.1 | 16 | 10000 | 64 | 0.001 | $10^{-7}$ | 30000 | 4 | $10^{-6}$ | 128 | 3000 | 16 | 0.001 |
| **Objects-Grass** | $\infty$ | 1 | 3000 | 64 | 0.001 | $10^{-6}$ | 30000 | 16 | $10^{-5}$ | 128 | 100000 | 64 | 0.0001 |
| | 10 | 1 | 3000 | 256 | 0.001 | $10^{-6}$ | 30000 | 16 | $10^{-5}$ | 32 | 3000 | 16 | 0.001 |
| | 3 | 1 | 3000 | 256 | 0.001 | $10^{-7}$ | 3000 | 4 | $10^{-5}$ | 32 | 3000 | 64 | 0.001 |
| | 1 | 1 | 3000 | 64 | 0.001 | $10^{-7}$ | 3000 | 4 | $10^{-5}$ | 32 | 3000 | 64 | 0.001 |
| | 0.1 | 4 | 10000 | 64 | 0.001 | $10^{-7}$ | 30000 | 16 | $10^{-6}$ | 128 | 3000 | 16 | 0.001 |
| **Objects-Autumn** | $\infty$ | 1 | 30000 | 64 | 0.0001 | $10^{-7}$ | 30000 | 16 | $10^{-6}$ | 128 | 100000 | 64 | 0.0001 |
| | 10 | 1 | 3000 | 64 | 0.001 | $10^{-6}$ | 30000 | 16 | $10^{-5}$ | 32 | 3000 | 64 | 0.001 |
| | 3 | 1 | 3000 | 256 | 0.001 | $10^{-7}$ | 3000 | 4 | $10^{-6}$ | 128 | 100000 | 4 | 0.0001 |
| | 1 | 1 | 3000 | 256 | 0.001 | $10^{-7}$ | 30000 | 16 | $10^{-5}$ | 32 | 3000 | 64 | 0.001 |
| | 0.1 | 16 | 10000 | 64 | 0.001 | $10^{-7}$ | 3000 | 4 | $10^{-6}$ | 32 | 100000 | 64 | 0.0001 |

Table 4: Optimal hyperparameter set-up for DP-DRE, DP-GAN-FT, DP-MEPF and DP-GAN-MI.

**DP-MEPF:** The learning rate $\eta$ is selected from $\{10^{-4}, 10^{-5}, 10^{-6}\}$ and the remaining settings follow the code released by Harder et al. (2022). During the training, the checkpoint is saved every 20,000 iterations. We save and present the *best* checkpoint along each training.

**DP-GAN-MI:** The private generator and discriminator both are 4-layer MLPs with latent dimension 25 and hidden width $w$ selected from $\{32, 128\}$. The learning rate $\eta$, the total number of training iterations $T$ and the batch size $B$ are selected from $\{10^{-3}, 10^{-4}\}$, $\{3 \times 10^3, 10^4, 10^5\}$ and $\{4, 16, 64\}$. The bounded norm $C$ is set as 1.0.

The evaluation results are based on the hyperparameter with the best FID score. Table 4 shows the exact hyperparameter set-up that is the optimal in the hyperparameter searching space.

# D ADDITIONAL EXPERIMENT RESULTS

## D.1 EVALUATION ON MORE DATASETS

In the main paper, we show the Precision and Recall and NDB evaluation results only on Cifar10 dataset. Table 5 and Figure 4 present the the results of these two evaluations on the remaining datasets. The results match the tendency of FID score (shown in the main paper): DP-GAN-MI sometimes does the best without privacy guarantee $\varepsilon = \infty$, while our two methods DP-MGE and DP-DRE are better than all baselines when $\varepsilon \leq 10$; Moreover, DP-MGE is comparable with DP-DRE when the dataset is unimodal such as bird and car, while DP-DRE is much better than DP-MGE when the dataset becomes more complicated.

| Method | Pet | | | | | | | | | |
|---|---|---|---|---|---|---|---|---|---|---|
| | $\varepsilon = \infty$ | | $\varepsilon = 10$ | | $\varepsilon = 3$ | | $\varepsilon = 1$ | | $\varepsilon = 0.1$ | |
| | Prec. | Rec. | Prec. | Rec. | Prec. | Rec. | Prec. | Rec. | Prec. | Rec. |
| Non-private IC-GAN | Precision: 0.880, Recall: 0.958 | | | | | | | | | |
| DP-GAN-FT | 0.797 | 0.854 | 0.000 | 0.000 | 0.425 | 0.487 | 0.354 | 0.448 | 0.388 | 0.461 |
| DP-MEPF | 0.444 | 0.425 | 0.568 | 0.566 | 0.510 | 0.574 | 0.452 | 0.361 | 0.013 | 0.000 |
| DP-GAN-MI | **0.888** | **0.951** | 0.255 | 0.181 | 0.092 | 0.083 | 0.080 | 0.117 | 0.060 | 0.127 |
| DP-MGE | 0.542 | 0.452 | 0.606 | 0.473 | 0.642 | 0.468 | 0.490 | 0.417 | 0.143 | 0.124 |
| DP-DRE | 0.870 | 0.921 | **0.853** | **0.905** | **0.874** | **0.912** | **0.867** | **0.919** | **0.491** | **0.613** |

| Method | Car | | | | | | | | | |
|---|---|---|---|---|---|---|---|---|---|---|
| | $\varepsilon = \infty$ | | $\varepsilon = 10$ | | $\varepsilon = 3$ | | $\varepsilon = 1$ | | $\varepsilon = 0.1$ | |
| | Prec. | Rec. | Prec. | Rec. | Prec. | Rec. | Prec. | Rec. | Prec. | Rec. |
| Non-private IC-GAN | Precision: 0.900, Recall: 0.948 | | | | | | | | | |
| DP-GAN-FT | 0.810 | 0.902 | 0.115 | 0.109 | 0.127 | 0.138 | 0.122 | 0.112 | 0.131 | 0.120 |
| DP-MEPF | 0.453 | 0.648 | 0.459 | 0.421 | 0.682 | 0.683 | 0.388 | 0.504 | 0.041 | 0.002 |
| DP-GAN-MI | 0.925 | **0.947** | 0.712 | 0.716 | 0.594 | 0.495 | 0.025 | 0.002 | 0.024 | 0.001 |
| DP-MGE | 0.903 | 0.855 | 0.899 | 0.907 | 0.889 | 0.868 | 0.739 | 0.783 | 0.041 | 0.021 |
| DP-DRE | **0.927** | 0.934 | **0.904** | **0.925** | **0.898** | **0.883** | **0.913** | **0.888** | **0.465** | **0.880** |

| Method | Bird | | | | | | | | | |
|---|---|---|---|---|---|---|---|---|---|---|
| | $\varepsilon = \infty$ | | $\varepsilon = 10$ | | $\varepsilon = 3$ | | $\varepsilon = 1$ | | $\varepsilon = 0.1$ | |
| | Prec. | Rec. | Prec. | Rec. | Prec. | Rec. | Prec. | Rec. | Prec. | Rec. |
| Non-private IC-GAN | Precision: 0.897, Recall: 0.946 | | | | | | | | | |
| DP-GAN-FT | 0.904 | 0.840 | 0.329 | 0.662 | 0.323 | 0.665 | 0.339 | 0.629 | 0.360 | 0.650 |
| DP-MEPF | 0.799 | 0.834 | 0.649 | 0.759 | 0.635 | 0.728 | 0.543 | 0.692 | 0.262 | 0.169 |
| DP-GAN-MI | **0.935** | 0.878 | 0.564 | 0.486 | 0.488 | 0.486 | 0.150 | 0.086 | 0.094 | 0.077 |
| DP-MGE | 0.928 | 0.707 | **0.940** | 0.712 | **0.919** | 0.744 | 0.720 | 0.736 | 0.230 | 0.167 |
| DP-DRE | 0.925 | **0.959** | 0.902 | **0.933** | 0.895 | **0.937** | **0.879** | **0.931** | **0.517** | **0.765** |

| Method | Objects-Grass | | | | | | | | | |
|---|---|---|---|---|---|---|---|---|---|---|
| | $\varepsilon = \infty$ | | $\varepsilon = 10$ | | $\varepsilon = 3$ | | $\varepsilon = 1$ | | $\varepsilon = 0.1$ | |
| | Prec. | Rec. | Prec. | Rec. | Prec. | Rec. | Prec. | Rec. | Prec. | Rec. |
| Non-private IC-GAN | Precision: 0.976, Recall: 0.970 | | | | | | | | | |
| DP-GAN-FT | 0.962 | 0.892 | 0.743 | 0.785 | 0.731 | 0.763 | 0.770 | 0.800 | 0.767 | 0.788 |
| DP-MEPF | 0.631 | 0.532 | 0.777 | 0.559 | 0.725 | 0.657 | 0.620 | 0.519 | 0.314 | 0.279 |
| DP-GAN-MI | 0.963 | 0.948 | 0.551 | 0.464 | 0.636 | 0.505 | 0.627 | 0.517 | 0.171 | 0.291 |
| DP-MGE | 0.791 | 0.682 | 0.844 | 0.756 | 0.859 | 0.759 | 0.855 | 0.767 | 0.602 | 0.624 |
| DP-DRE | **0.967** | **0.967** | **0.968** | **0.952** | **0.969** | **0.956** | **0.955** | **0.949** | **0.950** | **0.946** |

| Method | Objects-Autumn | | | | | | | | | |
|---|---|---|---|---|---|---|---|---|---|---|
| | $\varepsilon = \infty$ | | $\varepsilon = 10$ | | $\varepsilon = 3$ | | $\varepsilon = 1$ | | $\varepsilon = 0.1$ | |
| | Prec. | Rec. | Prec. | Rec. | Prec. | Rec. | Prec. | Rec. | Prec. | Rec. |
| Non-private IC-GAN | Precision: 0.967, Recall: 0.956 | | | | | | | | | |
| DP-GAN-FT | 0.958 | 0.896 | 0.798 | 0.717 | 0.820 | 0.714 | 0.752 | 0.723 | 0.767 | 0.720 |
| DP-MEPF | 0.668 | 0.668 | 0.747 | 0.737 | 0.549 | 0.542 | 0.269 | 0.179 | 0.116 | 0.062 |
| DP-GAN-MI | **0.956** | **0.955** | 0.679 | 0.457 | 0.318 | 0.294 | 0.378 | 0.354 | 0.157 | 0.312 |
| DP-MGE | 0.753 | 0.643 | 0.783 | 0.687 | 0.819 | 0.732 | 0.804 | 0.736 | 0.409 | 0.632 |
| DP-DRE | 0.930 | 0.923 | **0.938** | **0.898** | **0.936** | **0.908** | **0.941** | **0.884** | **0.910** | **0.849** |

Table 5: Precision and recall (higher is better) of our methods and baselines on Pet, Bird, Car, Objects-Grass and Objects-Autumn.

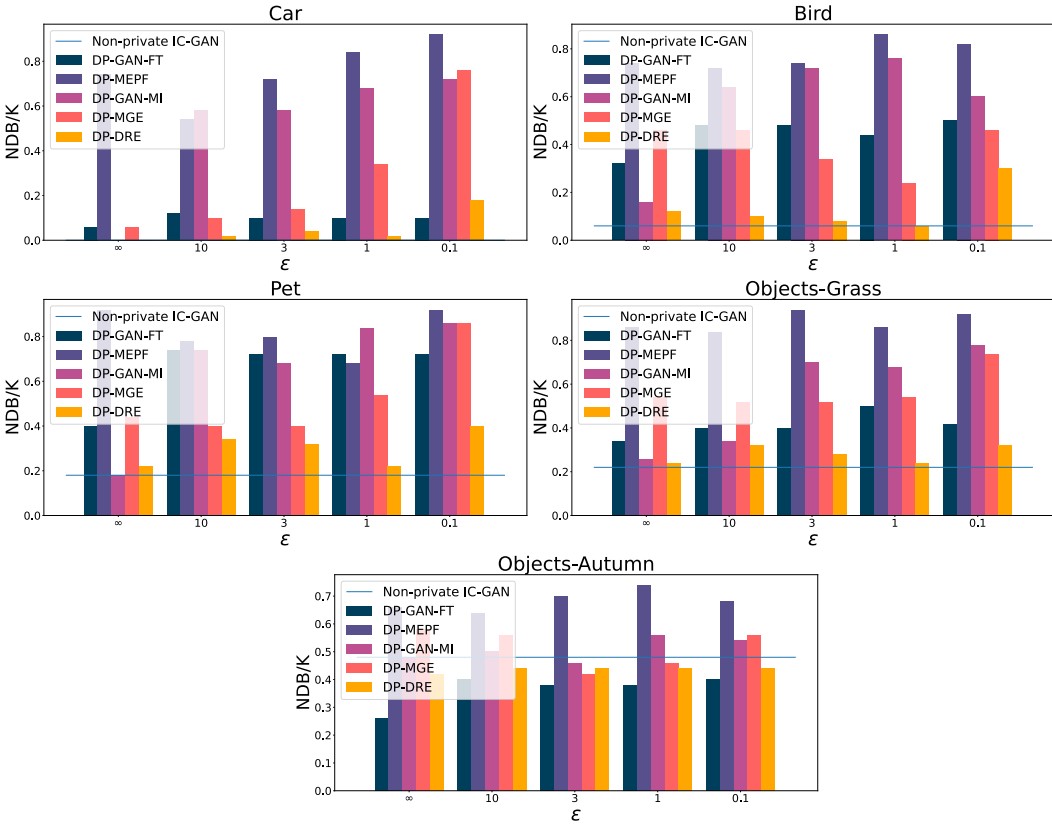

Figure 4: The percentage of different bins NDB/K (lower is better) of our methods and baselines on Pet, Bird, Car, Objects-Grass and Objects-Autumn.

## D.2 GENERATED EXAMPLES

In the main paper, we show the examples generated from different algorithms when $\varepsilon = 1$. Figure 5 Figure 6, Figure 7, and Figure 8 show the generation results for $\varepsilon = \infty, 10, 3, 0.1$.

By checking these examples, we find that when $\varepsilon = 1$ (in Figure 2 in the main paper), DP-MGE and DP-DRE still generate related objects (dataset Pet, Car, Bird) or contexts (Objects-Grass), but they fail when $\varepsilon = 0.1$. This trade-off is better than the baselines. Two GAN related baselines DP-GAN-FT and DP-GAN-MI are not capable to generate in-distribution images for some datasets when $\varepsilon \leq 10$. DP-MEPF generates good images on Cifar10 but images with many artifacts on the remaining high-resolution dataset.

One observation for DP-DRE is that it doesn't generate very autumn-like images for the Objects-Autumn dataset even if $\varepsilon = \infty$. We hypothesize that the reason would be the feature extractor doesn't perfectly capture the autumn features, because Non-private IC-GAN fails to generate autumn images as well.

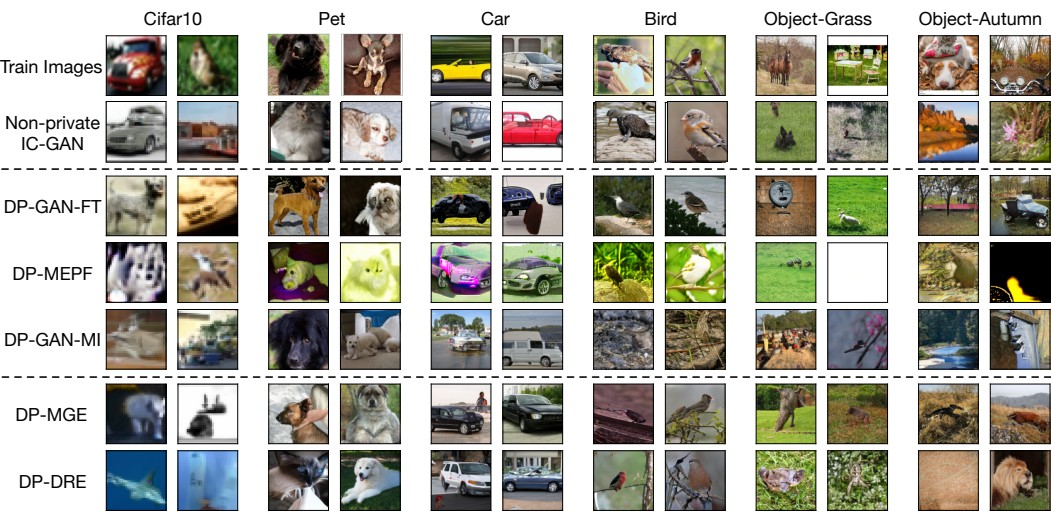

Figure 5: Examples of all algorithms across six datasets when $\varepsilon = \infty$.

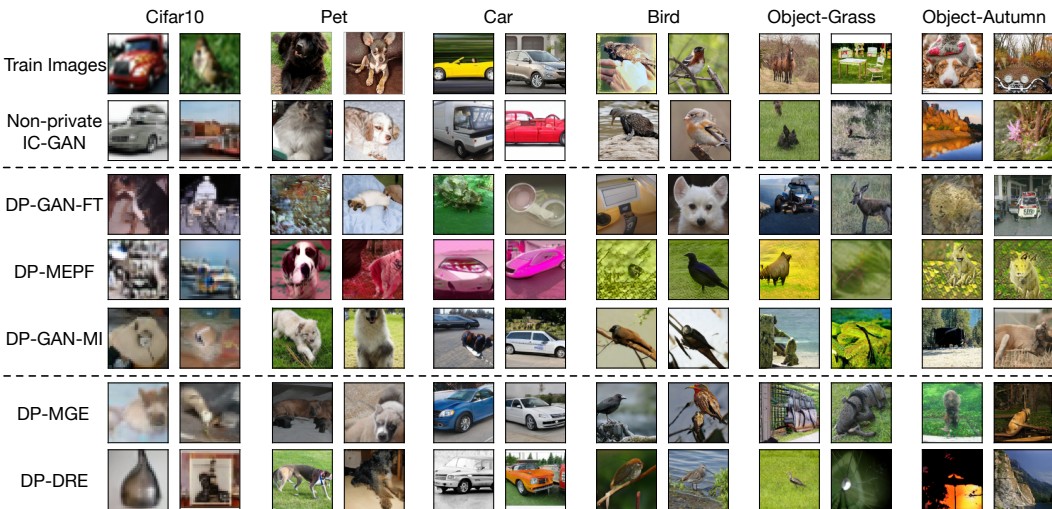

Figure 6: Examples of all algorithms across six datasets when $\varepsilon = 10$.

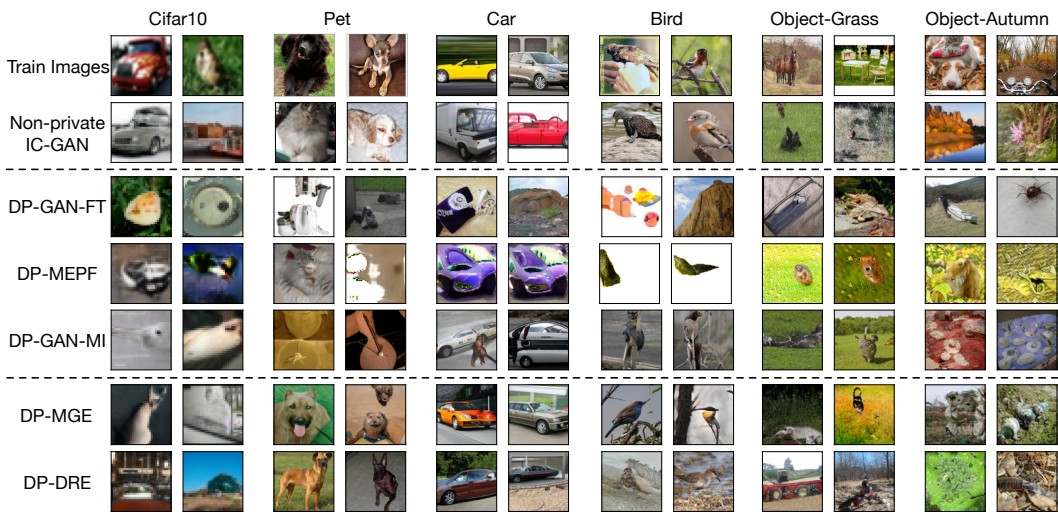

Figure 7: Examples of all algorithms across six datasets when $\varepsilon = 3$.

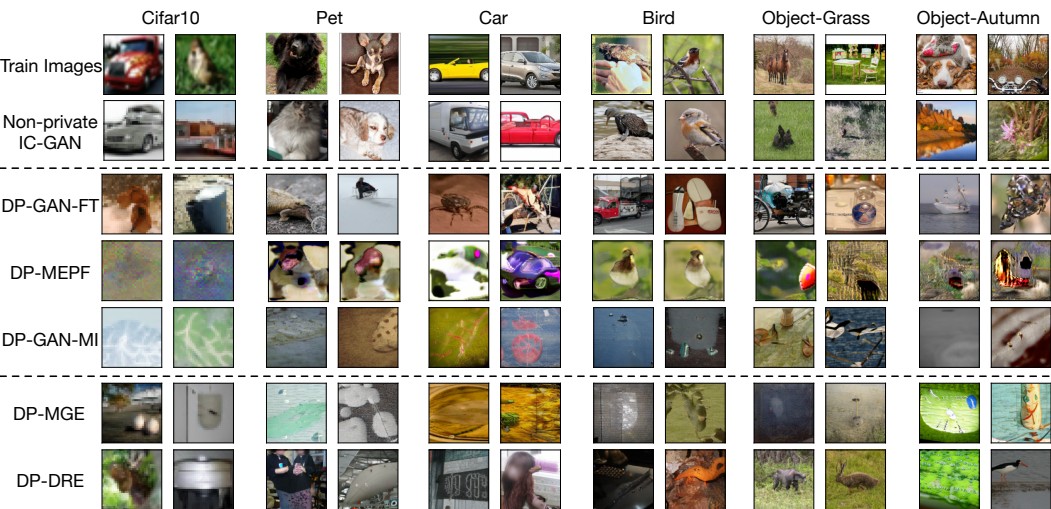

Figure 8: Examples of all algorithms across six datasets when $\varepsilon = 0.1$.

