# OpenReview forum: "Large-Scale Public Data Improves Differentially Private Image Generation Quality"
_ICLR.cc/2024/Conference — Submitted to ICLR 2024_

### Official Review · Reviewer_3Hzn · 2023-10-24

**Soundness:** 4 excellent
**Presentation:** 3 good
**Contribution:** 3 good
**Rating:** 6
**Confidence:** 5

**Summary:**

-	The paper proposes a private generation method by assuming the support of the public data, which contains the private dataset and can be assumed as the Imagenet dataset for the public data. Then, the authors introduce DP-MGE and DP-DRE, where DP-DRE demonstrates prominent performance in private generation using the IC-GAN architecture.

**Strengths:**

-	The paper exhibits its own novelty in designing the density ratio estimator rather than fine-tuning the pre-trained models trained with public data. The latter is widely used in private generation tasks involving public data recently.
-	By using the discriminator to distinguish whether the latent vector $\boldsymbol{v}$ is obtained from public or private sources, the authors identify the optimal density ratio with a simple optimal assumption and convert the latent information from public data to private data with straightforward multiplication.
-	The paper demonstrates the performance improvement of DP-DRE across a wide range of datasets, outperforming existing methods. The authors include various tasks such as the FID score, precision, and recall.
-	The paper is well-written and easy to read.

**Weaknesses:**

-	Please refer to questions.

**Questions:**

I will happily raise the score if the authors can address the following questions:

1.	I agree the discriminator method is very interesting. However, considering that conditional generation is important, how can you inject conditioning into your framework? This is my major concern.

2.	Even though the authors mentioned the weakness of DP-MGE and highlighted the strength of DP-DRE in the comparison results in Section 4, I'm somewhat curious whether the performance of DP-MGE is influenced by the large amount of noise in $\mu$ and $\sigma$ in the latent dimension or the assumption of a unimodal Gaussian approach. Considering that conventional GANs use sampling starting from the unimodal normal distribution, the unimodal assumption itself might not be that harmful. Instead, the difference during training, such as injecting noise into $\mu$ and $\sigma$ versus DP-SGD training of the discriminator, could be a reason. (Minor) Alternatively, the authors might consider exploring complex distributions such as a mixture of Gaussians (MoG) for private latent space.

3.	(Minor) Can you explain how to ensure $||v_i|| \leq 1$ in the normalization operator in IC-GAN? Does the operator clip the norm to 1 or use other methods to ensure the norm?

4.	I agree that using public data in terms of the discriminator is different from existing approaches. However, one of my concerns is that recent approaches in DP synthesis emphasize the importance of DP-SGD rather than existing methods [1,2], which is also true for classification tasks [3] using larger network sizes. One of my concerns is that DP-GAN-FT may be too naïve to test the performance of DP-DRE. As you mentioned, adopting fine-tuning is easy for existing methods, as you discussed in Section 5. Can you provide additional experimental results using state-of-the-art (SOTA) architectures or optimizations without using public data, on top of pre-trained models?

5.	Although the proposed method focuses on GANs, there are various existing models that use pre-trained models in private diffusion synthesis. As they use pre-trained models in the same setting as the proposed methods, the authors should address these topics [4,5].

6.	There are some typos, such as two neibouring datasets in Appendix A. Please re-check the grammar once again.

[1] Differentially Private Diffusion Models, TMLR 2023.
[2] Private GANs, TMLR 2023.
[3] Unlocking High-Accuracy Differentially Private Image Classification through Scale, Arxiv 2022.
[4] Differentially Private Latent Diffusion Models, Arxiv 2023.
[5] Differentially Private Diffusion Models Generate Useful Synthetic Images, Arxiv 2023.

**Details Of Ethics Concerns:**

The paper deals with the major situation of differentially private deep learning. Thus, the authors should provide the code of ethics.

---

> ### Author Response · Authors · 2023-11-18
> **Officient Comment by Submission446 Authors**
>
> Thank you for your positive and insightful feedback to our paper! We will add the discussions below in the revision.
>
> **Conditional generation**: Thank you for pointing out the issue of conditional generation. We could extend our method by applying our method for each class while sharing a joint privacy budget; this is what DP-MEPF does for conditional generation. Although we believe that our advantage in image quality would apply in the conditional generation setting as well, we think advanced adaptation to conditional generation would be worth studying as future work.
>
> **The limitation of DP-MGE**: We believe that the limitation of DP-MGE is because it models the private distribution in the feature space as a unimodal Gaussian, and not because of the noise added due to DP. This is because we found that in the $\varepsilon = \infty$ case, DP-MGE did quite poorly in some datasets (Pet, Objects-Grass, Objects-Autumn) in our experiments. We will clarify and discuss this in the paper.
>
> Using a mixture of a small number of Gaussians (gaussian mixture models) instead of a single Gaussian is also a good idea; However, we think DP-DRE might still outperform this method for low privacy budgets ($\varepsilon\leq 1$). Notice that increasing the complexity of distribution representation (e.g. the number of Gaussian) will boost the performance when $\varepsilon=\infty$, this algorithm will become less robust to the noise due to DP than the unimodal Gaussian. For example, the accuracy DP-MGE significantly drops when $\varepsilon=1.0$ on the Pet dataset, which implies gaussian mixture models would significantly drop when $\varepsilon=1.0$ as well. By comparison, DP-DRE when $\varepsilon=1.0$ is capable of mostly achieving the same performance as non-private IC-GAN on Pet dataset – which is an upper bound on all IC-GAN based algorithms. Same analysis can be applied into other datasets.
>
> **The normalization operator in IC-GAN**: This is designed inside the IC-GAN. Once the feature extractor (ResNet50) $h’$ is trained, the training procedure of the decoder (GAN) always uses the normalized features $h(x_i):=h’(x_i)/\|h’(x_i)\|$ from ResNet50.
>
> **Comparison with SOTA private training diffusion models**: Thank you for pointing out the references to the diffusion models! We will add the discussion and comparisons below to our paper.
> [1] considers the more challenging pure private data/no pre-trained model setting. Consequently, their results apply to image datasets with low-resolution (MNIST, etc.)
> [2, 3] study DP fine-tuning on a publicly pre-trained model, where the public data is ImageNet – the same as our set-up. As their code does not appear to be publicly available, we will compare performances on CIFAR10, which is reported in their paper. For CIFAR10 – although [2,3] have lower FID scores when $\varepsilon \geq 5.0$, our method DP-DRE starts to outperform [2, 3] when $\varepsilon=1.0$ – DP-DRE has FID score of $20.9$ while [2, 3] have FID scores of $22.9$ and $25.2$. Indeed our method is also tested with $\varepsilon=0.1$ and its FID score still resists at $20.9$! This demonstrates the advantage of our method at the low $\varepsilon$ (high privacy) regime, while their method may do better at high $\varepsilon$ (low privacy) regimes.
>
> **Relevant topics and typos**: Thanks for pointing out those relevant work and typos. We will address them in the revision.
>
> [1] Differentially Private Diffusion Models, TMLR 2023.
>
> [2] Differentially Private Latent Diffusion Models, Arxiv 2023.
>
> [3] Differentially Private Diffusion Models Generate Useful Synthetic Images, Arxiv 2023.

---

> ### Comment · Reviewer_3Hzn · 2023-11-22
> **Rebuttal follow-up**
>
> I have carefully reviewed the authors' rebuttal and the feedback from other reviewers. I thank for the authors for their comprehensive response and the additional effort they have put into enhancing the manuscript. Specifically, I appreciate the authors to include recent works on private image generation and the almost of my concerns are clarified.
>
> However, as I mentioned in Question1, I think it is hard to inject the conditional generation framework into DP-DRE since the class-wise mapping using IC-GAN is trained with bigger models (i.e., ImageNet) and corresponding class labels cannot be found on private domain such as CIFAR. Can you give me the idea more specifically about them? From my perspective, conditional generation is not selective works for generation tasks, but essential parts, especially for privacy.
>
> Therefore, I will maintain my current score for the manuscript and await the comments of other reviewers.

---

### Official Review · Reviewer_KC1E · 2023-10-30

**Soundness:** 3 good
**Presentation:** 3 good
**Contribution:** 2 fair
**Rating:** 5
**Confidence:** 4

**Summary:**

This paper studies differentially private image generation under the assumtion that large-scale public data is available. More specifically, the public data assumption posits that the support of the public data distribution contains the support of the private data. This work incorporates the IC-GAN and reduces the problem of learning the private image distribution to learning a private feature distribution in the latent space of the encoder. This work proposes two methods DP-MGE and DP-DRE, for differentially private embedding estimation. The experiments evaluation includes 6 different image tasks and achieves compelling results across different privacy levels.

**Strengths:**

- The evaluation is comprehensive, across different datasets and different privacy levels and achieves compelling results.
- The paper is well-written and easy to follow.

**Weaknesses:**

- It seems to me that the most significant improvements by DP-DRE/DP-MGE stem from the use of IC-GAN. For example, for DP-DRE, even at epsilon=0, i.e., by directly using the public data embedding, the results are already much better than most private results by DP-GAN-FT/DP-MEPF/DP-GAN-MI. Also, the DP-MGE looks similar to DP-MEPF. Both of these methods estimate the first and second order statistics. According to the non-private results, the improvement for DP-MGE over DP-MEPF is that it would be better to directly use the embedding instead of training another network to approximate such embedding.

- Another concern I have is regarding the assumption of public data. This work assumes that the support of the public data distribution contains the support of the private data and the authors validate that DP-DRE/DP-MGE perform poorly when the assumption is severely violated. However, recent works show that even out-of-distribution public data can still help private training [R1]. It would be better to study how to adapt to the private embedding domain when this assumption is violated.

[R1] Arun Ganesh, Mahdi Haghifam, Milad Nasr, Sewoong Oh, Thomas Steinke, Om Thakkar, Abhradeep Guha Thakurta, Lun Wang. Why Is Public Pretraining Necessary for Private Model Training? ICML 2023

**Questions:**

- In Table 2, it is somewhat counterintuitive that the private result of DP-MGE at epsilon =1 is better than the result of DP-MGE at epsilon=infinity. I wonder if there is a brief explanation for this.
- If I understand correctly, DP-GAN-FT and DP-GAN-MI share the same architecture but the trainable parameters are different. While DP-GAN-FT has more trainable parameters, for the non-private results, DP-GAN-MI performs better. Does this indicate that the pretrained features in IC-GAN matter most and updating it using the private data would distort the pretrained features? This might indicate that DP-MGE/DP-DRE improves by the advantage of IC-GAN, while facing the issue that cannot adapt to the private domain when the public data assumption is violated.

---

> ### Author Response · Authors · 2023-11-18
> **Officient Comment by Submission446 Authors**
>
> Thank you for your insightful feedback to our paper! We will include the discussions below in the revision.
>
> **Comparison between DP-MGE and DP-MEPF**: This is actually an insightful point! It is correct that both DP-MGE and DP-MEPF estimate the first and second order statistics. DP-MGE directly uses the embedding while DP-MEPF instead approximates such embedding. However, DP-MGE has the help of IC-GAN to generate images from given features, while DP-MEPF has to train a generator from scratch to match the statistics. DP-MGE outperforming DP-MEPF actually demonstrates the advantage of adopting the IC-GAN framework.
>
> **The justification of the assumption**: We agree that especially motivated by [R1] how to leverage out-of-distribution public data for the task of data generation is one important future direction. Alternatively, regarding our assumption that the support of the public data distribution covers the support of the private, with time, this assumption is becoming increasingly true – as foundation models get trained on larger and larger datasets covering larger and larger domains.
>
> **Explanation for the results of DP-MGE**: Because there is a large gap between DP-MGE and Non-private IC-GAN, this implies that the unimodal Gaussian with the correct mean and variance is very different from the true private distribution in the feature space. Because the mean and variance of the private distribution is not necessary to be the optimal mean and variance in the unimodal Gaussian to describe the private distribution, a small amount of additive noise to the mean and variance estimation is possible to improve the performance.
>
> **Comparison between DP-GAN-FT and DP-GAN-MI**: This is a very insightful question! DP-GAN-MI is to train a GAN in the feature space, where the image information is already extracted by the pretrained feature mapping. However, DP-GAN-FT is training a GAN in the image space and has to learn more implicit connection between the image space and the latent space, which is harder by intuition.

---

> > ### Comment · Reviewer_KC1E · 2023-11-19
> > **Thank you for your response**
> >
> > Thank you for your response. I find the explanation of Table 2 solves my Q1. I still have concerns for the significance of this work regarding W1 and W2 listed above (Q2 is also somewhat related to W1/W2). Therefore I maintain my score.

---

> > > ### Author Response · Authors · 2023-11-20
> > > **Further explanation for the key advantage of IC-GAN**
> > >
> > > We appreciate your quick response and find the Q1 has been solved! With regards to the W1, we would like to provide more insights about how DP-DRE deeply leverages the advantage of IC-GAN rather than a straight extension of IC-GAN. From these insight, we will see the overall good performance of DP-DRE reply on both the role of IC-GAN and, more importantly, how DP-DRE is designed on top of IC-GAN. We hope this can address your concern on the novelty/significance of the method.
> > >
> > > DP-DRE benefits from two key ideas and IC-GAN plays a crucial role for the performance of DP-DRE, which makes IC-GAN distinguished from other generative models:
> > > 1. Converting the generation task from image space to feature space.
> > >     - IC-GAN is such an auto-encoder to make this conversion and it provides a time-efficient way to compute both directions (image$\to$feature, feature$\to$image). By comparison, image$\to$feature in a standard GAN, e.g. stylegan, is well-known as GAN-inversion, which is hard in terms of time complexity and the performance [1].
> > > 2. Learning the distribution in a discriminative way to better leverage the public data. Unlike other methods which learn the private distribution from scratch (in the feature space, e.g. DP-GAN-MI), DP-DRE only needs to learn a boundary to select representative data from public data if public data has a subset well representing the private data.
> > >     - A good latent space can make this boundary easier to learn. The latent space in IC-GAN is a self-supervised-learning feature space and both public and private data distribution have good structures in such a feature space. This reason especially makes IC-GAN distinguished from other auto-encoder or generative models. By comparison, for example, the public distribution in the latent space of GAN and VAE is a normal distribution and has less structure to distinguish it from the private data.
> > >
> > > [1] Gan inversion: A survey. IEEE Transactions on Pattern Analysis and Machine Intelligence 45.3 (2022): 3121-3138.

---

### Official Review · Reviewer_Kyfi · 2023-10-30

**Soundness:** 3 good
**Presentation:** 2 fair
**Contribution:** 2 fair
**Rating:** 5
**Confidence:** 2

**Summary:**

In this paper, the authors studied the differential privacy (DP) image generation. The author's goal is to train a Generative Adversarial Network (GAN) to generate samples from the private data distribution while preserving differential privacy of the private data. I am not expert of DP image generation. However, I am concerned about the following issues.

**Strengths:**

The authors' proposed algorithm outperforms existing benchmark aspects of FID scores as well as other distribution quality metrics.

**Weaknesses:**

Lack of comparison of state-of-the-art comparison methods.

**Questions:**

Major comments:
1.	Please add the latest comparison methodology (e.g., for 2023).
2.	In subsection 4.2, “These artifacts may be due to the fact that DP-MEPF does not use a pretrained public encoder.”, In subsection 4.2, "A" is described, is there any experiment added to prove it. For example, artifacts can be eliminated by adding pre-trained public coding to DP-MEPF.
3.	The paper can be compared with more adversarial training-based generative models, such as the few-shot migration methods such as Diffusion-GAN, to demonstrate the advantages of the methodology.
Minor comments：
1.	There are grammatical problems in the paper. It is suggested to check and modify it carefully. For example:
1)	In the abstract, “while the private data consist of images of a specific type”->” while the private data consists of images of a specific type”.
2)	In subsection 2.1, “and now considered”->”and is now considered”
3)	And so on.
2.	Subsection 5, related work, is proposed to be placed after introduction.
3.	Please add a reference to the comparison method DP-GAN-FT.

---

> ### Author Response · Authors · 2023-11-18
> **Officient Comment by Submission446 Authors**
>
> Thank you for your constructive feedback to our paper!
>
> **Comparison with SOTA private training diffusion models**: Since the review doesn’t specify the latest methodology, we follow the suggestion from Reviewer 3Hzn and compare our method with [1, 2, 3].
> [1] considers the more challenging pure private data/no pre-trained model setting. Consequently, their results apply to image datasets with low-resolution (MNIST, etc.)
> [2, 3] study DP fine-tuning on a publicly pre-trained model, where the public data is ImageNet – the same as our set-up. As their code does not appear to be publicly available, we will compare performances on CIFAR10, which is reported in their paper. For CIFAR10 – although [2,3] have lower FID scores when $\varepsilon \geq 5.0$, our method DP-DRE starts to outperform [2, 3] when $\varepsilon=1.0$ – DP-DRE has FID score of $20.9$ while [2, 3] have FID scores of $22.9$ and $25.2$. Indeed our method is also tested with $\varepsilon=0.1$ and its FID score still resists at $20.9$! This demonstrates the advantage of our method at the low $\varepsilon$ (high privacy) regime, while their method may do better at high $\varepsilon$ (low privacy) regimes.
>
> **The explanation for DP-MEPF**: This indeed can be validated by the comparison between DP-MGE and DP-MEPF. Both DP-MGE and DP-MEPF estimate the first and second order statistics, while DP-MGE further utilizes a pretrained decoder in IC-GAN to generate images from the statistics and DP-MEPF trains a generator from scratch to match the statistics. We can observe that DP-MGE is able to generate realistic high resolution images.
>
> **Comparison with Diffusion-GAN**: As our paper focuses on the privacy-preserving image generation, instead of the effectiveness of the generation model, the hardness is mainly to achieve a great privacy-utility tradeoff (how to inject noise to make it private while keeping a good performance). We have discussed in the related work section that our method has better privacy-utility tradeoff than GAN-based methods with differential privacy in the literature.
>
> [1] Differentially Private Diffusion Models, TMLR 2023.
>
> [2] Differentially Private Latent Diffusion Models, Arxiv 2023.
>
> [3] Differentially Private Diffusion Models Generate Useful Synthetic Images, Arxiv 2023.

---

### Official Review · Reviewer_VNLs · 2023-10-31

**Soundness:** 3 good
**Presentation:** 3 good
**Contribution:** 2 fair
**Rating:** 5
**Confidence:** 4

**Summary:**

This submission investigates the task of differentially private (DP) image generation while leveraging public data. Specifically, an "autoencoder" (a feature extractor that encodes images into a feature space, combined with an ICGAN generator that produces images from these features) is pre-trained on large-scale public datasets such as ImageNet.

To achieve DP generation, the authors first estimate the feature distribution of the private data using existing DP density estimation methods. Samples are then drawn from this estimated distribution and fed into the pre-trained decoder (i.e., the ICGAN generator).
DP distribution estimation is realized through either:
- A DP Gaussian estimation, termed the DP-MGE variant.
- Training an auxiliary GAN (that approximately acts as a density ratio estimator) in the feature space, termed the DP-DRE variant.

Experimental results demonstrate that the proposed method generally surpasses several baseline methods in generation quality, including DP-GAN-FT, DP-Merf, and DP-GAN-MI.

**Strengths:**

- The paper is generally well-written and easy to follow
- Utilizing public data for DP generation is a natural idea for improving privacy-utility trade-off
- The key assumption (i.e., the support of the public data distribution contains the support of the private one) is clearly stated
- Experiments generally demonstrate promising results

**Weaknesses:**

- The assumption is a bit strong (i.e., the private distribution should be fully covered by the public one) which may greatly restrict the applicability of the proposed approach. While this limitation has been acknowledged by the authors in section 4.2, a potential adaptation or extension of the method to fit the common practical scenario is currently missing but would greatly strengthen the submission.

- This work's originality seems to offer little innovation due to its dependence on two established components: the use of public data (numerous references) and the implementation of an autoencoder-style pipeline (refer to [1,2]). Both of these strategies have previously been investigated for DP generation.

- The experimental comparison appears to overlook some essential baselines or references, making the current work less convincing. For example, given that it is quite a natural idea to use foundation models for DP learning, a critical baseline would be using strong pre-trained generative models (such as strong GANs like StyleGAN2 and Diffusion models [3]) and directly fine-tune such models or filter the outputs.


[1] "Differentially private data generative models", Arxiv 2018
[2] "Differentially private mixture of generative neural networks", IEEE Transactions on Knowledge and Data Engineering, 2018
[3] "Differentially Private Diffusion Models Generate Useful Synthetic Images", Arxiv, 2023

**Questions:**

- As discussed above, using external public data for DP generation/learning seems both intuitive and promising. However, I would have expected the paper to delve deeper into why the proposed usage stands out as superior, especially when there are straightforward alternatives that leverage powerful foundational models. While I'm not aware of published work that focuses on this idea, the significance of this paper's contribution may still be diminished if no clear advantages are highlighted in comparison to these straightforward options. This is particularly true since I've observed superior results from other options when dealing with distribution shifts, and such shifts are inevitable when using public data.

- From what I understand, private generation in the proposed approach simply requires sampling of features that follow the private distribution, rather than a full density estimation. Given this, why wouldn't it be more effective to train the GAN to generate samples directly, rather than using an approximate density estimation for the DP-DRE variant? Especially considering that GANs don't natively support the density estimation and it could lead to inaccurate results.

- I feel there's a lack of context regarding how ICGAN operates and why it's particularly suitable. A concise description of its mechanism and a few sentences highlighting its key advantages would be beneficial.

---

> ### Author Response · Authors · 2023-11-18
> **Officient Comment by Submission446 Authors**
>
> Thank you for your constructive feedback!
>
> **The justification of the assumption**: Regarding our assumption that the support of the public data distribution covers the support of the private, with time, this assumption is becoming increasingly true – as foundation models get trained on larger and larger datasets covering larger and larger domains. However, note that this does not mean that there is no distribution shift – as the private distribution may assign a lot of probability mass to parts of the space where the public distribution has very low (but still non-zero) mass. An example is when the public data are common objects from all over the world, and private data (e.g. DollarStreat dataset) are the objects from developing countries.
>
> **Contributions**: While the use of public data and an autoencoder-style pipeline have been investigated in isolation, our main contribution is to find the right algorithmic technique that can obtain good results – namely, the use of IC-GAN to convert the generation problem in the image space into the feature space, and careful design of the distribution learning method in the feature space leveraging the public data again (in the next paragraph, we will explain more about the key advantages of IC-GAN). We agree that the final algorithm is simple, but in this case, simplicity is a strength as it leads to good private performance!
>
> **The key advantages of IC-GAN**: Thank you for asking this insightful question. DP-DRE has two design ideas and IC-GAN plays a crucial role for the performance of DP-DRE:
> 1. Converting the generation task from image space to feature space.
>     - IC-GAN is such an auto-encoder to make this conversion and it provides a time-efficient way to compute both directions (image$\to$feature, feature$\to$image). By comparison, image$\to$feature in a standard GAN, e.g. stylegan, is well-known as GAN-inversion, which is hard in terms of time complexity and the performance [1].
> 2. Learning the distribution in a discriminative way to better leverage the public data. Unlike other methods which learn the private distribution from scratch (in the feature space, e.g. DP-GAN-MI), DP-DRE only needs to learn a boundary to select representative data from public data if public data has a subset well representing the private data.
>     - A good latent space can make this boundary easier to learn. The latent space in IC-GAN is a self-supervised-learning feature space and both public and private data distribution have good structures in such a feature space. This reason especially makes IC-GAN distinguished from other auto-encoder or generative models. By comparison, for example, the public distribution in the latent space of GAN and VAE is a normal distribution and has less structure to distinguish it from the private data.
>
> [1] Gan inversion: A survey. IEEE Transactions on Pattern Analysis and Machine Intelligence 45.3 (2022): 3121-3138.

---

> ### Author Response · Authors · 2023-11-18
> **Officient Comment by Submission446 Authors**
>
> Thank you for proposing comparison with meaningful baseline, which helps better demonstrate the advantage of our method empirically.
>
> **Comparison with the usage of the foundation model**: Filtering out the outputs from a strong foundation model. Indeed, we make this attempt as described in the paragraph “Rejection sampling in the latent space versus the image space” in Section 4.2. IC-GAN is one of the SOTA image generation methods as well and we implement the rejection sampling on the IC-GAN pretrained on ImageNet. Also, notice that DP-DRE is essentially the rejection sampling in the feature space. As stated in the paragraph “Rejection … space”, the DP-DRE has better performance and less computation cost.
>
> **Comparing DP-DRE with training a GAN in the feature space**: Our baseline DP-GAN-MI is to learn a GAN in the feature space and the empirical results of DP-GAN-MI are much worse than DP-DRE in the private setting. Actually, GAN is to learn a distribution from scratch, while DP-DRE, leveraging the public data, only needs to learn a boundary to select representative data from public data and is not necessary to learn what features should look like. The latter process can be very robust to the noise introduced from the DP algorithm especially when the representative data has a certain margin with the remaining data.
>
> **Comparing with advanced other architecture-based methods**: Thanks for raising this point. We agree that having a paragraph that summarizes the comparison with other architecture-based methods can help understand the position of our method in the literature and will add the discussion below to our experiment section.
> GAN-based DP algorithm. As stated in the related work, the GAN-based DP algorithm in the literature is far away from generating realistic high resolution images at the low $\varepsilon$ regime; they are working towards MNIST.
> Diffusion-based DP algorithm.
> 1. [1] considers the more challenging pure private data/no pre-trained model setting. Consequently, their results apply to image datasets with low-resolution (MNIST, etc.)
> 2. [2, 3] study DP fine-tuning on a publicly pre-trained model, where the public data is ImageNet – the same as our set-up. As their code does not appear to be publicly available, we will compare performances on CIFAR10, which is reported in their paper. For CIFAR10 – although [2,3] have lower FID scores when $\varepsilon \geq 5.0$, our method DP-DRE starts to outperform [2, 3] when $\varepsilon=1.0$ – DP-DRE has FID score of $20.9$ while [2, 3] have FID scores of $22.9$ and $25.2$. Indeed our method is also tested with $\varepsilon=0.1$ and its FID score still resists at $20.9$! This demonstrates the advantage of our method at the low $\varepsilon$ (high privacy) regime, while their method may do better at high $\varepsilon$ (low privacy) regimes.
>
> [1] Differentially Private Diffusion Models, TMLR 2023.
>
> [2] Differentially Private Latent Diffusion Models, Arxiv 2023.
>
> [3] Differentially Private Diffusion Models Generate Useful Synthetic Images, Arxiv 2023.

---

> > ### Comment · Reviewer_VNLs · 2023-11-23
> >
> > Dear authors,
> >
> > Thank you for your rebuttal and for providing additional results. I believe that a more comprehensive comparison, including the use of recent powerful foundation models such as diffusion models, is necessary. This would allow for a more detailed assessment of the outcomes in the suggested setting, as the current version of the submission does not seem to fully reflect the additional experiments.
> >
> > In its current form, if the paper's primary claim is "our main contribution is to find the right algorithmic technique that can obtain good results'', this seems somewhat narrow in scope, especially when contrasted with the broader implications suggested by the paper's title. Moreover, the extent of experiments and depth of insights presented are, in my view, marginally below the standard expected for an ICLR publication. Thus, I decide to keep my score and will defer the final decision to the discussion with other reviewers and the AC.

---

### Meta-Review · Area_Chair_2aQY · 2023-12-11

**Metareview:**

The reviewers generally appreciate the clarity and methodology of the paper, particularly its use of public data for differentially private data generation and its performance across various datasets. However, there were concerns about the strong assumptions, which assume that the public data contains the support of private data. The reviewers also suggest there is inadequate comparison with state-of-the-art techniques. The reviewers made a few suggestions, including more comprehensive comparisons, exploration of conditional generation, and addressing the influence of relaxing the assumptions. They also highlight minor issues such as grammatical errors and suggest improvements for future work.

**Justification For Why Not Higher Score:**

The idea of using public data for differentially private data generation has been around in the literature for a few years now. This paper adds another idea for this approach. However, the reviewers identified several issues about the execution of the results (listed above).

**Justification For Why Not Lower Score:**

N/A

---

### Decision · Program_Chairs · 2024-01-16

Reject